



# The Centrifugal Differential Mobility Analyser - A new device for determination of two-dimensional property distributions

Torben N. Rüther[1], David B. Rasche[1,2], and Hans-Joachim Schmid[1]

[1]Particle Technology Group, Mechanical Engineering, Paderborn University, Paderborn, 33098, Germany
[2]REMBE GmbH Safety+Control, Brilon, 59929, Germany

**Correspondence:** Hans-Joachim Schmid (hans-joachim.schmid@uni-paderborn.de)

**Abstract.** Usually, for the characterisation of nanoparticles an equivalent property is measured, e.g. the mobility equivalent diameter. In the case of non-spherical, complex shaped nanoparticles, one equivalent particle size is not sufficient for a complete characterisation. Most of the methods utilised to gain deeper insight into the morphology of nanoparticles are very time consuming and costly or have bad statistics (such as tandem-setups or SEM/TEM images). To overcome these disadvantages, a prototype of a new compact device, the Centrifugal Differential Mobility Analyser, the CDMA, was built, which can measure the full two-dimensional distribution of mobility and stokes equivalent diameters by classification in a cylinder gap, through electrical and centrifugal forces. An evaluation method to determine the transfer probabilities is developed and used in this work to compare the measurement results with the theory for the pure rotational behaviour (like the Aerodynamic Aerosol Classifier) and the pure electrical behaviour (like the Dynamic Mobility Analyser). In addition, the ideal two-dimensional transfer function was derived using a particle trajectory approach. This two-dimensional transfer function is a prerequisite for obtaining the full two-dimensional particle size distribution from measurements by inversion.

## 1 Introduction

In the world of nanoparticles, there exist numerous ways to characterise them. A common feature of many techniques is that, for non-spherical particles, they measure an equivalent particle size - eg. the Differential Mobility Analyser-DMA, (Knutson and Whitby, 1975) measures the hydrodynamic equivalent diameter; the Aerodynamic Aerosol Classifier-AAC, (Tavakoli and Olfert, 2013), the Low Pressure Impactor-LPI, (Fernandez de la Mora et al., 1989) and the Aerodynamic Particle Sizer-APS, (Mitchell et al., 2003) measure the aerodynamic equivalent diameter; the Centrifugal Particle Mass Analyser-CPMA, (Olfert and Collings, 2005) measures the mass-charge-ratio. However, this information alone is not sufficient to determine the actual particle properties comprehensively, especially in the case of large agglomerates, which may have a significantly different surface area than volume equivalent spherical particles. Therefore, numerous studies have focused on this topic. In particular, the influence of particle shape on bio-availability and toxicity (Jindal, 2017; Toy et al., 2014) as well as on environmental aerosols, represents a topic of high scientific interest (Kelesidis et al., 2022). Increased reaction rates due to the higher surface area and the effect of particle shape on the mechanical stability of batteries have also been investigated (Zhang et al., 2022).



Due to its relevance, many research institutions are focusing on separation by more than one particle property to produce highly specific particle systems (Rhein et al., 2019; Sandmann and Fritsching, 2023; Furat et al., 2020, 2019).

In order to design and control these processes effectively, it is essential to develop techniques to assess particle structure and dimensions. SEM studies at least offer full shape information in 2D for this purpose, but imaging the full particle size distribution requires a significant number of images, which can be both time consuming and costly. An alternative approach are tandem setups, i.e. a serial arrangement of two different measurement systems, allowing the determination of two different equivalent particle sizes. This can provide more comprehensive information and can be used to derive enhanced, more specific structure information, e.g. the effective density or fractal dimension (Park et al., 2008; Slowik et al., 2004; Tavakoli and Olfert, 2014).

The CDMA (Centrifugal Differential Mobility Analyser) is a recently developed compact device that has been designed to address the limitations of existing tandem setups, particularly the high costs of equipment and the complexity of the associated measurement procedures. Furthermore, the measurement and evaluation can be conducted directly with the CDMA, which significantly reduces the analytical burden and the user's required knowledge calculating such measurements.

The objective is to obtain a complete two-dimensional particle size distribution expressed in terms of the stokes and mobility equivalent diameter. The combination of these two properties allows us to draw conclusions on the particle geometry, i.e. a complete two-dimensional property distribution of the effective density or fractal dimension can be calculated, thus facilitating an even more precise and comprehensive investigation of the distribution shape. Additionally, the large number of examined particles enhances the statistical reliability of the findings, exceeding SEM examinations. Furthermore, additional examinations can be conducted using this approach. For example, it may be possible to measure the charge distribution of specific particles or to investigate the influence of particle shape on charge distribution.

## 2 Concept and fundamental theory

The newly developed principle of CDMA combines the concepts of DMA and AAC. In general, the CDMA consists of two concentric cylinders between which high voltage can be applied, and/or both of which can be rotated at the same angular speed. This means that both the voltage and the speed can be superimposed, whereas in the DMA, only the voltage can be varied, and in the AAC, only the speed. This means that with the CDMA, particles can be classified according to their mobility and stokes equivalent diameter. By measuring all voltage-speed combinations, a full two-dimensional particle size distribution in terms of $d_{st}$ and $d_m$ can be calculated by data inversion.

When an aerosol volume flow $Q_a$ is applied to the inner cylinder, the particles are displaced by electrical forces and centrifugal forces which always drive the particles toward the outer cylinder (Fig. 1). Particles pass through the sheath air flow $Q_{sh}$ and are classified and counted in the sample flow $Q_s$ if they meet the specified characteristics.

Because inertial forces are typically negligible, the quasi-static particle drift velocity $w_{Dr}$ can be obtained from a force balance. Assuming Stokes' drag then leads to:

$$Q_P \cdot E + m_P \cdot a_c = 3\pi \eta d_m w_{Dr}/Cu \qquad (1)$$





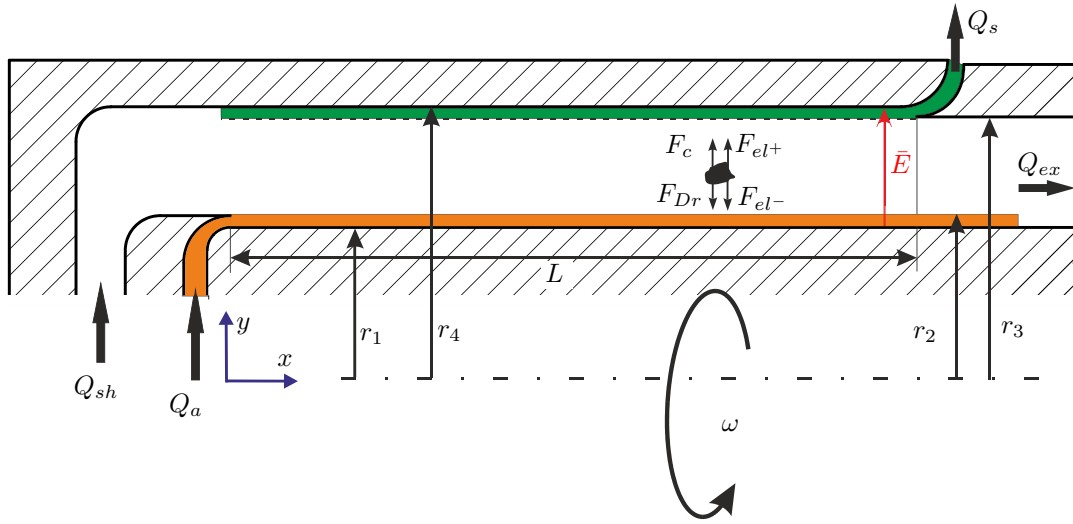

**Figure 1.** Schematic drawing of the classification zone of a CDMA: sheath air $Q_{sh}$, exhaust air $Q_{ex}$, sample air $Q_s$, aerosol air $Q_a$, electric field magnitude $E$, rotational speed $\omega$, electrical force $F_{el}$, centrifugal force $F_c$, drag force $F_{el}$, length of the transfer path $L$, inner radius $r_1$, maximum radius at which the particles enter $r_2$, minimum radius of which the particles are still classified $r_3$ and outer radius $r_4$.

With particle charge $Q_P$, electric field magnitude $E$, particle mass $m_P$, centrifugal acceleration $a_c = \omega^2 r$, dynamic viscosity $\eta$, mobility diameter $d_m$, particle drift velocity $w_{Dr}$, Cunningham correction factor $Cu$ (Allen and Raabe, 1985).

The limiting cases $E = 0$ or $\omega = 0$, equation (1) leads to the equations from the derivations of the DMA (Stolzenburg, 1988)
60 and AAC (Tavakoli and Olfert, 2014), respectively.

Using the same assumptions as in the boundary cases - i.e. no diffusion, plug flow, no perturbations of the E-field (ideal geometry/no room-charges, no inertial forces) - the deterministic description of the particle's path is achieved by rearranging and integrating equation (1).

$$r\left(y\right) = \sqrt{\frac{\left(\tau \cdot \omega^2 \cdot r_{in}^2 + \frac{Z \cdot U}{\ln\left(\frac{r_4}{r_1}\right)}\right) \cdot \exp\left\{2 \cdot \tau \cdot \omega^2 \cdot y \cdot \frac{\pi \cdot \left(r_4^2 - r_1^2\right)}{Q_{sh} + Q_a}\right\} - \frac{Z \cdot U}{\ln\left(\frac{r_4}{r_1}\right)}}{\tau \cdot \omega^2}} \tag{2}$$

65 With voltage $U$, rotational velocity $\omega$, inner $r_4$ and outer $r_1$ radius, actual radius at which the particle enters $r_{in}$, length of the classifying gap $L$, position of the particle in streamwise direction $y$, particle relaxation time $\tau$ (Tavakoli and Olfert, 2014)[1] and particle mobility $Z$ (Stolzenburg, 1988).

$$\tau = \frac{\rho \cdot d_v^3 \cdot Cu(d_m)}{18 \cdot d_m \cdot \eta} = \frac{\rho \cdot d_{st}^2 \cdot Cu(d_m)}{18\eta} \tag{3}$$

---

[1]See small comment for the definition of the particle relaxation time in Appendix A





$$Z = \frac{n \cdot e \cdot Cu(d_m)}{3\pi \cdot \eta \cdot d_m} \tag{4}$$

Where $\rho$ is the particle density, $d_{st}$ is the stokes equivalent diameter, $d_v$ is the volume equivalent diameter, $n$ is the number of charges carried by a particle, $e$ is the elementary charge and $Q_{ex}$ is the exhaust gas volume flow.

The dimensionless, normalised mobility or particle relaxation time is obtained as follows:

$$\widetilde{Z} = Z/Z^* \quad ; \quad \widetilde{\tau} = \tau/\tau^* \tag{5}$$

Where $Z^*$ is the mobility required for a particle entering at the centre of the aerosol inlet to be sampled exactly at the centre of the outlet (Stolzenburg, 1988). $\tau^*$ describes the same behaviour, but for the relaxation time (Tavakoli and Olfert, 2014).

$$Z^* = \frac{Q_{sh} + Q_{ex}}{4 \cdot \pi L U} \cdot \ln\left(\frac{r_4}{r_1}\right) \tag{6}$$

$$\tau^* = \frac{Q_{sh} + Q_{ex}}{\pi \omega^2 (r_1 + r_4)^2 L} \tag{7}$$

For typical operating conditions ($Q_s = Q_a$), a dimensionless form of equation (2) can be derived, where $\bar{r} = \frac{r_1 + r_4}{2}$ is the average of the outer and inner radii, $\beta = \frac{Q_a}{Q_{sh}}$ is the ratio of aerosol to sheath airflow, and $\widetilde{h} = \frac{r_4 - r_1}{\bar{r}}$ is the ratio of gap height to $\bar{r}$.

$$r(y) = \sqrt{r_{\text{in}}^2 \cdot \exp\left\{\frac{y}{L} \cdot \widetilde{\tau} \cdot \frac{2\widetilde{h}}{1+\beta}\right\} + \bar{r}^2 \cdot \frac{\widetilde{Z}}{\widetilde{\tau}} \cdot \left[\exp\left\{\frac{y}{L} \cdot \widetilde{\tau} \cdot \frac{2\widetilde{h}}{1+\beta}\right\} - 1\right]} \tag{8}$$

## 3 CDMA Prototype

To validate the functional principle, a prototype was designed and built. As a boundary condition, this prototype should be able to measure particle sizes ranging from 50 nm to 1000 nm for both the mobility equivalent diameter and the stokes equivalent diameter. In addition, the speed should not exceed 6000 rpm, because the ferrofluidic sealing has only been tested in this range (with higher angular speeds the sealing could evaporate much faster and produce particles itself) and to prevent unbalanced forces on the bearings.

### 3.1 Design

Figure 2 shows the cross section of the CDMA prototype and its dimensions.

With these properties (and assuming a particle density of 1 kg/m$^3$), it is possible to characterise particles in the size range from 50 nm to 1000 nm where the maximum speed is 6000 rpm and the maximum voltage is limited to 1000 V$^2$.

---

[2]A maximum of 3 kV/mm gap distance can be applied with optimally dry air and smooth flat surfaces [4]. As there are corners, particularly at the inlet and outlet, a maximum voltage of 300 V/mm is chosen for safety reasons.

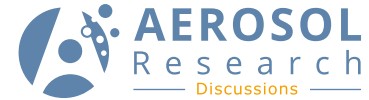

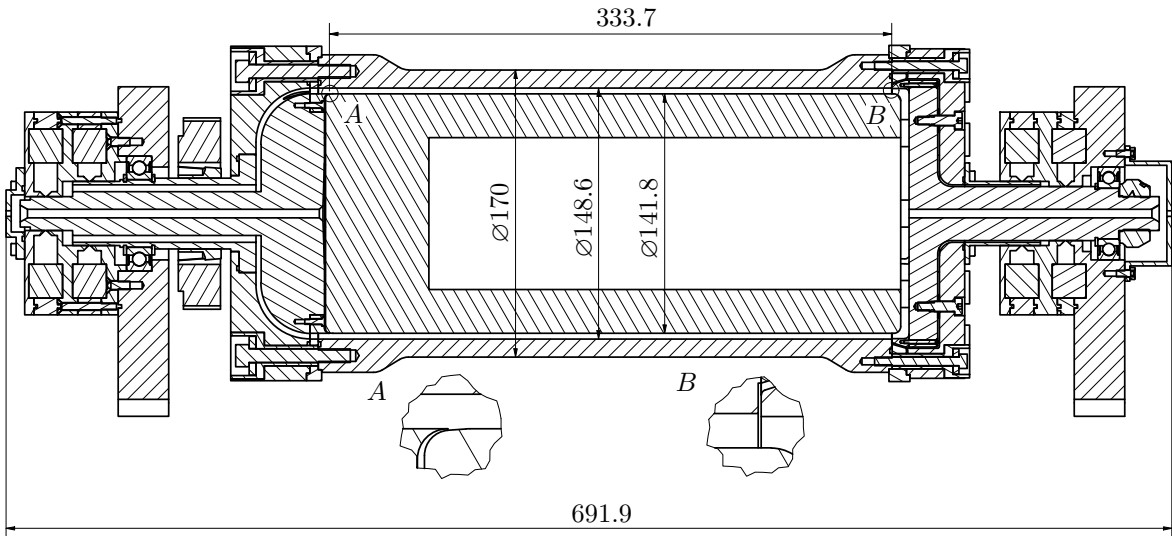

**Figure 2.** Cross section of the CDMA prototype

For both, the mobility and the stokes equivalent diameter, one difficulty was to achieve a suitable seal against the environment.

Friction seals are not an option, as either the sealing of the system is difficult or the seal heats up strongly due to high friction, generates particles or abrasion occurs in general. Therefore, a ferrofluid seal was designed and tested. A ring magnet is used. On the outside of the ring magnet, there are iron components that create a pole shoe on the inside of the ring. This pole shoe has a tolerance of approximately +0.1 mm to the shaft passing through it. The ferrofluid is injected into the pole shoe. This creates a virtually frictionless seal that does not generate particles.


The aerosol is fed into the long bore on the left, entering the classifying gap at point A. The sheath air is fed between the two ferrofluidic sealings (this drilling is not visible in fig. 2), entering the CDMA through 8 axial holes and finally also the classifying gap via a bend. At the end of the classifying gap (position B), the sample flow is diverted outwards so that the particles are directed through narrow gaps toward the ferrofluidic sealing and released at the centre of the seal. The sampled

particles can then be counted by a CPC or similar instrument. The exess gas flow is sucked in toward the centre and directed through the long bore to the right, where it is purified to be returned to the inlet of the CDMA as the sheath air flow.

A toothed belt is used to transmit the forces of the motor to the rotating cylinder. The high voltage is applied to the centre of the outer cylinder. This area is electrically isolated from the rest of the CDMA by insulators (outside between points A and B). The other components are connected to earth.

## 3.2 Particle losses

Particle losses occur due to the classification principle in the inlet and outlet areas of the CDMA. This occurs, in particular, during rotation, as the centrifugal force then acts on the particles in all rotating feed and discharge pipes. This movement pushes





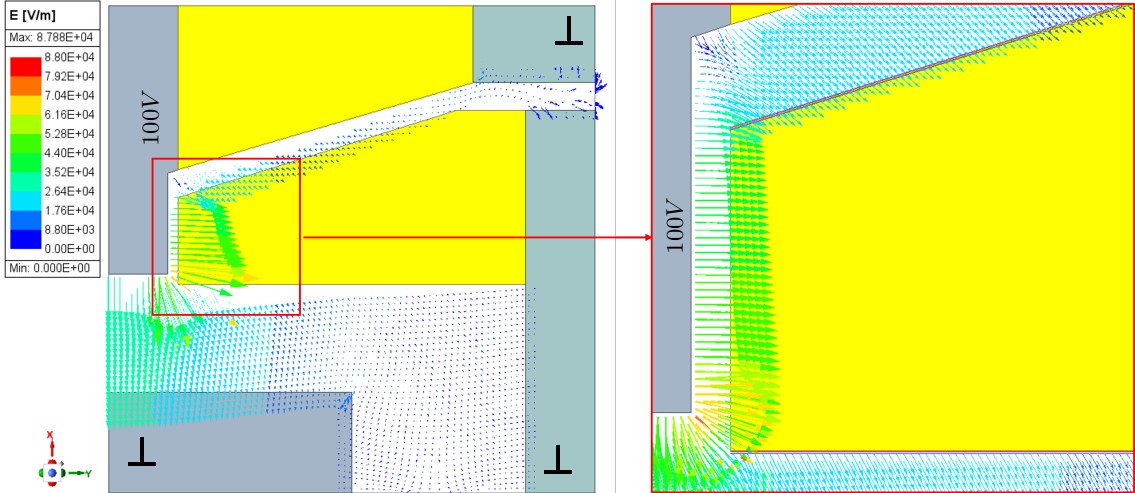

**Figure 3.** Electrostatic simulation of the outlet using 'Ansys Electronics' at an applied voltage of 100 V

the particles further toward the respective outer wall, where they are separated. Walls with a large radius running parallel to the axis of rotation are particularly susceptible because of the higher exerting centrifugal forces.

Separation can be calculated individually for each part. Dividing the radial distance $s$ the particles move in each part by the distance between the walls $s_{max}$ gives the degree of separation, assuming a laminar flow profile and uniform concentration across the flow cross section[3].

$$T = s/s_{max} \tag{9}$$

In addition, losses also occur when the voltage is applied. Since a polymeric material (yellow regions in Fig. 3) was used to
insulate the high voltage outer electrode directly at the input and output region, an electric field is even generated in the input and output gap due to the insulating properties of the air. Figure 3 shows a simulation with the Ansys Electronics software, which was used to simulate the field strengths at an applied voltage of 100 V. It can be seen that the field strength is partly as high as in the classifying gap. Hence, the simulation can also be used to calculate a theoretical deposition analogous to rotation.

**4  The Transfer function**

The transfer function $\Omega$ describes the probability that a particle with certain properties (relaxation time $\tau$ and mobility $Z$) will be successfully classified under given operating conditions (voltage $U$ and angular velocity $\omega$).

---

[3]This only affects measured number concentration and therefore only the maximum height of the transfer function as neither the width nor the position is affected $\Omega_{max} = \frac{\Omega_{max,measured}}{(1-T)}$.



### 4.1 Two-dimensional transfer function based on the particle trajectory calculation

Since in CDMA, as in DMA and AAC, the ratio of aerosol volume flow to sheath air volume flow cannot be infinitely small

and the inlet and sample gaps are also finite, the classified aerosol is not completely monomodal, but a distribution exists. Assuming a constant particle flux density at the inlet, stratified flow, and homogeneous E-field in the classifying gap, as well as non-inertial and diffusion-free particles, this distribution can be calculated analytically. The derivation of the two-dimensional transfer function is given in Appendix B.

Therefore, the transfer function $\Omega$ can be calculated as follows:

$$\Omega_{CDMA} = \max\left[\min\left(f_1, f_2, 1\right), 0\right] \tag{10}$$

With:

$$f_1 = \frac{\frac{\beta+\kappa^2}{1+\beta} - \left(\frac{1/\beta+\kappa^2}{1/\beta+1} + \frac{(\kappa+1)^2}{4} \cdot \frac{\widetilde{Z}}{\widetilde{\tau}}\right) \cdot \exp\left\{-\widetilde{\tau} \cdot \frac{2\widetilde{h}}{1+\beta}\right\} + \frac{(\kappa+1)^2}{4} \cdot \frac{\widetilde{Z}}{\widetilde{\tau}}}{\frac{\beta+\kappa^2}{1+\beta} - \kappa^2} - \ldots$$

$$\ldots \max\left(\frac{\frac{\beta+\kappa^2}{1+\beta} - \left(1 + \frac{(\kappa+1)^2}{4} \cdot \frac{\widetilde{Z}}{\widetilde{\tau}}\right) \cdot \exp\left\{-\widetilde{\tau} \cdot \frac{2\widetilde{h}}{1+\beta}\right\} + \frac{(\kappa+1)^2}{4} \cdot \frac{\widetilde{Z}}{\widetilde{\tau}}}{\frac{\beta+\kappa^2}{1+\beta} - \kappa^2}, 0\right) \tag{11}$$

$$f_2 = \frac{\left(1 + \frac{(\kappa+1)^2}{4} \cdot \frac{\widetilde{Z}}{\widetilde{\tau}}\right) \cdot \exp\left\{-\widetilde{\tau} \cdot \frac{2\widetilde{h}}{1+\beta}\right\} - \frac{(\kappa+1)^2}{4} \cdot \frac{\widetilde{Z}}{\widetilde{\tau}} - \kappa^2}{\frac{\beta+\kappa^2}{1+\beta} - \kappa^2} - \ldots$$

$$\ldots \max\left(\frac{\left(\frac{1/\beta+\kappa^2}{1/\beta+1} + \frac{(\kappa+1)^2}{4} \cdot \frac{\widetilde{Z}}{\widetilde{\tau}}\right) \cdot \exp\left\{-\widetilde{\tau} \cdot \frac{2\widetilde{h}}{1+\beta}\right\} - \frac{(\kappa+1)^2}{4} \cdot \frac{\widetilde{Z}}{\widetilde{\tau}} - \kappa^2}{\frac{\beta+\kappa^2}{1+\beta} - \kappa^2}, 0\right) \tag{12}$$


$$\kappa = \frac{r_1}{r_4} = \frac{1 - \widetilde{h}/2}{1 + \widetilde{h}/2} \tag{13}$$

$$\beta = Q_a / Q_{sh} \tag{14}$$

### 4.2 Theoretical Transfer functions for $\widetilde{\tau} = 0$ and $\widetilde{Z} = 0$

Figure 4 presents the transfer functions for $\widetilde{\tau} = 0$ and $\widetilde{Z} = 0$ of the CDMA, respectively. For $\widetilde{\tau} = 0$ (Fig. 4a), the transfer function becomes that of a normal DMA. This is indicated by the typical triangular shape, where the FWHM (full width at half maximum) value also corresponds to the value for beta. It can also be seen that there is no dependence on the slit geometry, because the curves for both $\widetilde{h} = 0.05$ and $\widetilde{h} = 0.5$ are identical.

The transfer function for $\widetilde{Z} = 0$ (Fig. 4b) exhibits pure AAC behaviour. In contrast to the traditional AAC-theory (Tavakoli and





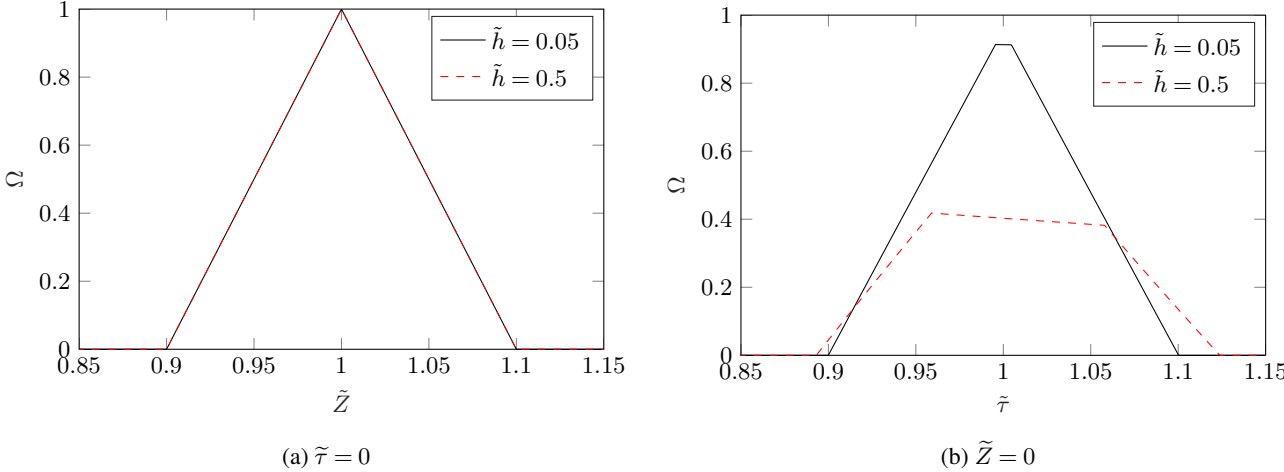

**Figure 4.** Transfer functions for $\widetilde{\tau} = 0$ and $\widetilde{Z} = 0$ at $\beta = 0.1$ for different ratios of radii.

Olfert, 2014), the transfer function has not a triangular shape. This is due the assumption of a mean centrifugal force acting on the particles throughout the whole classification gap, which has been used thus far. However, as described in the previous section, increased particle deposition occurs because the centrifugal force increases with increasing radius. This means that if particles with $\widetilde{\tau} = 1$ are fed over the inlet, all particle streamlines should only be shifted parallel. However, as the centrifugal force increases with radius, particles entering the classifier gap at a larger radii, are directly affected by a higher centrifugal

force. Thus, if the particles are close to $r_2$, they already have a larger radius when they enter the transfer zone and therefore a higher centrifugal force. Figure 5a shows individual particle trajectories (with $\widetilde{\tau} = 1$ and $\widetilde{Z} = 0$) for particles fed between $r_1$ and $r_2$ (blue dashed line) and sampled between $r_3$ and $r_4$ (the 2 red dashed lines) after transfer length $L$.

It can be seen that the particle trajectories are widened to such an extent that particles are deposited both before and after the sampling gap - streamlines which do not end between the red lines at $L$ are deposited on the walls before or after the sampling

outlet. This phenomenon is, of course, due to the way in which the AAC works and therefore occurs mainly when the particle relaxation time is relevant. This is also why the ideal transfer function for an AAC is a truncated triangular function. This increases as the $\widetilde{h}$ value increases, so the shape of the transfer function becomes increasingly distorted and declining values for $\Omega$.

### 4.3   Theoretical Two-dimensional transfer function

Equation (10) can be used to calculate the transfer probability for each combination of $\widetilde{\tau}$ and $\widetilde{Z}$. If $\beta = 0.1$ and $\widetilde{h} = 0.05$, the two-dimensional transfer function shown in Fig. 5b is obtained. Here, the influence of the widening streamlines increases with decreasing values of $\widetilde{Z}$ and increasing values of $\widetilde{\tau}$ and the height of transfer function decreases. It should be noted that, in contrast to $\widetilde{\tau}$, there are also negative values for $\widetilde{Z}$. This is due to the presence of both positively and negatively charged particles. For $\widetilde{Z} > 0$, the direction of the electric force on the particles coincides with the centrifugal force. For $\widetilde{Z} < 0$ it acts



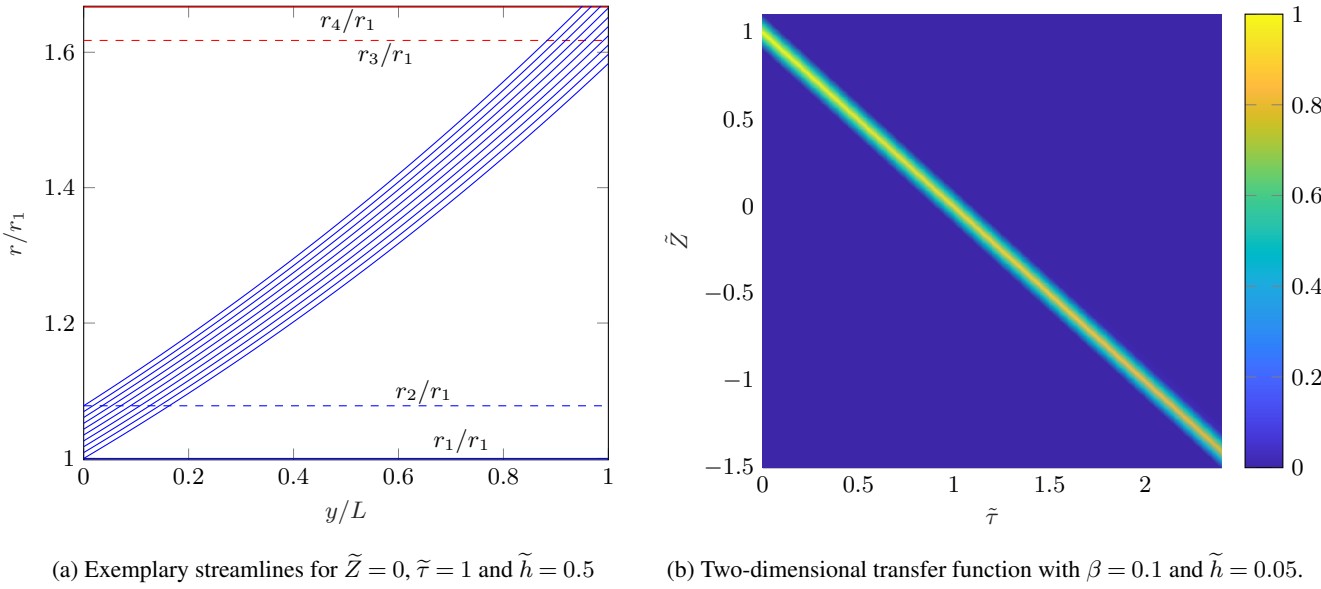

(a) Exemplary streamlines for $\widetilde{Z} = 0$, $\widetilde{\tau} = 1$ and $\widetilde{h} = 0.5$      (b) Two-dimensional transfer function with $\beta = 0.1$ and $\widetilde{h} = 0.05$.

**Figure 5.** Exemplary Streamlines and two-dimensional transfer function

in the opposite direction. Using the 2-dimensional transfer function, the classification probabilities can be calculated for each operating point, providing a matrix for data inversion, which is required for back-calculation.

## 5   Measurement of the transfer functions for the DMA and AAC operating modes

To validate the instrument functionality, the transfer functions of the two transfer functions for $\widetilde{\tau} = 0$ and $\widetilde{Z} = 0$ (DMA mode i.e. $\omega = 0$ /AAC mode i.e. $U = 0$) should first be determined experimentally. To measure a DMA/AAC transfer function, a

tandem setup consisting of two instruments was used, where the first instrument continuously provides a mono-mobile aerosol, while the second instrument scanned the whole measuring range step by step.

In this case, a classifier (TSI 3080) with a DMA (TSI 3081) was used as the previous instrument. The voltage and desired volume flows are set there. The second device was the CDMA, to which the same volume flows are applied via another classifier (TSI 3080). To measure the transfer function, the measuring range of the CDMA is scanned step by step, i.e. the

voltage/speed is increased discretely and the resulting concentration $n_2$ is measured. As the concentration after the first unit ($n_1$) is also important for the calculation, the second unit can be bypassed using 2 valves. For the aerosol production, two tube furnaces and an agglomeration tube are used. Figure 6 shows a scheme of this complete setup.

### 5.1   Production of a test aerosol

To obtain meaningful measurements, it is also important to produce a stable and constant test aerosol. This is achieved by

heating silver to $1150°C$ in a hot wall reactor. During this process, some of the silver evaporates into the gas phase. When the





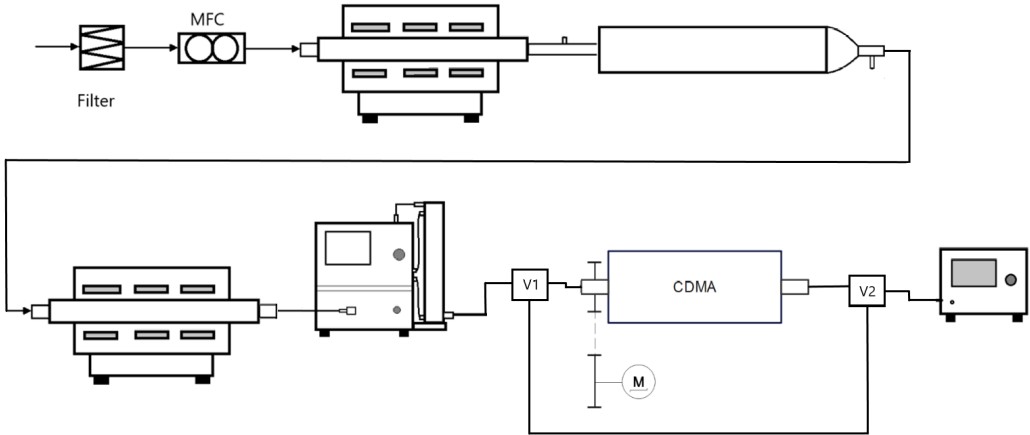

**Figure 6.** Schematic of the entire experimental setup: Test aerosol production with two tubefurnaces (Nabertherm) and an agglomeration tube and the consecutive setup consisting of a classifier (TSI 3080) with a DMA (TSI 3081), CDMA and CPC (TSI 3775) for the measurement of a transfer function

temperature is lowered at the outlet of the hot wall reactor, silver precipitates as nanoparticles. Larger agglomerates are formed in the agglomeration tube.

These agglomerates can be heated to approximately 700°C in a second hot wall reactor in order to thermally round them into spherical particles. Figure 7 shows the mean particle diameter of the test aerosol after agglomeration. It can be seen that mean particle size and total number concentration remains very stable over a long period of time. Because it takes approximately 30 minutes to measure a transfer function, it can be assumed that the particle size distribution is stable. Figure 8a shows the roundness of the particles and thus underscores their suitability for the first CDMA evaluation.

### 5.2 Calculating a transfer function from measurement data assuming a gaussian shape

If the tubing length from the first valve (V1) to the second instrument plus the tubing length from the second instrument to the second valve (V2) is equal to the length of the tubing, which is the bypass of the second instrument thus, the tubing loss term is eliminated (Li et al., 2006). The efficiency of the Condensation Particle Counter (CPC) is also eliminated if exactly the same CPC is used to measure $n_1$ and $n_2$ (see fig. 6 and section 5).

Since the first instrument is a DMA, the determination of the transfer function for the DMA-mode (i.e. $\omega = 0$) is done in the following. For the Quotient $n_2/n_1$ the following formula applies (Li et al., 2006)[4]:

$$n_2/n_1 = \frac{\int\limits_{-\infty}^{+\infty} \Omega_1\left(\widetilde{Z}\right) \cdot \Omega_2\left(\widetilde{Z}\right) \mathrm{d}\widetilde{Z}}{\int\limits_{-\infty}^{+\infty} \Omega_1\left(\widetilde{Z}\right) \mathrm{d}\widetilde{Z}} \tag{15}$$

---

[4]The dimensionless form of $n_2/n_1$ as given in equation (15) can be derived from the dimensioned equation given by *Li et al.*.

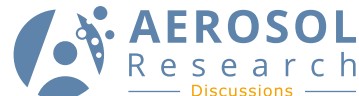

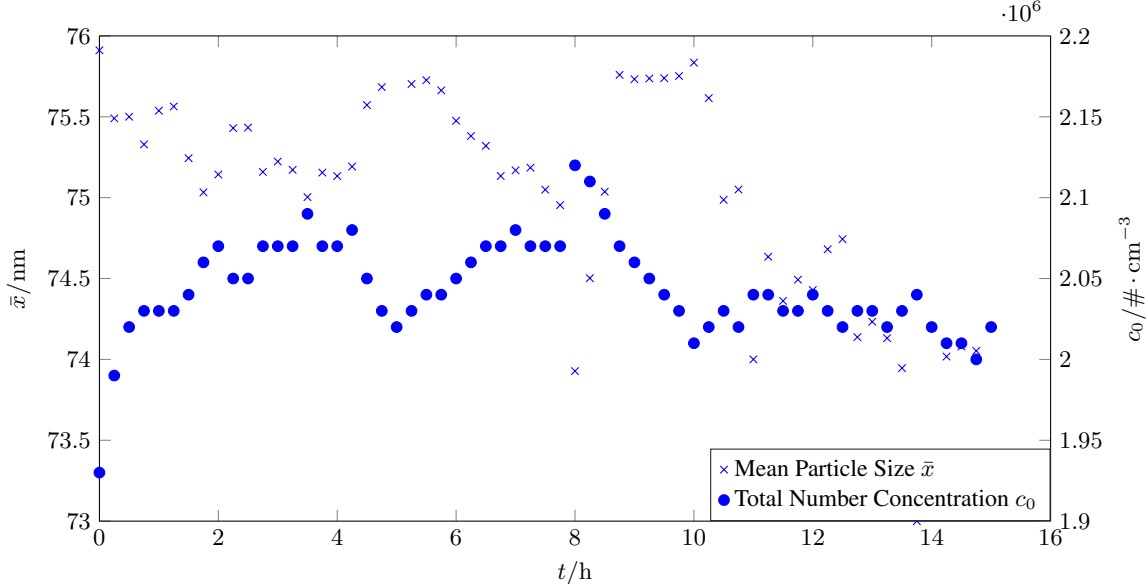

**Figure 7.** Mean particle diameter and total concentration measured by the SMPS over time.

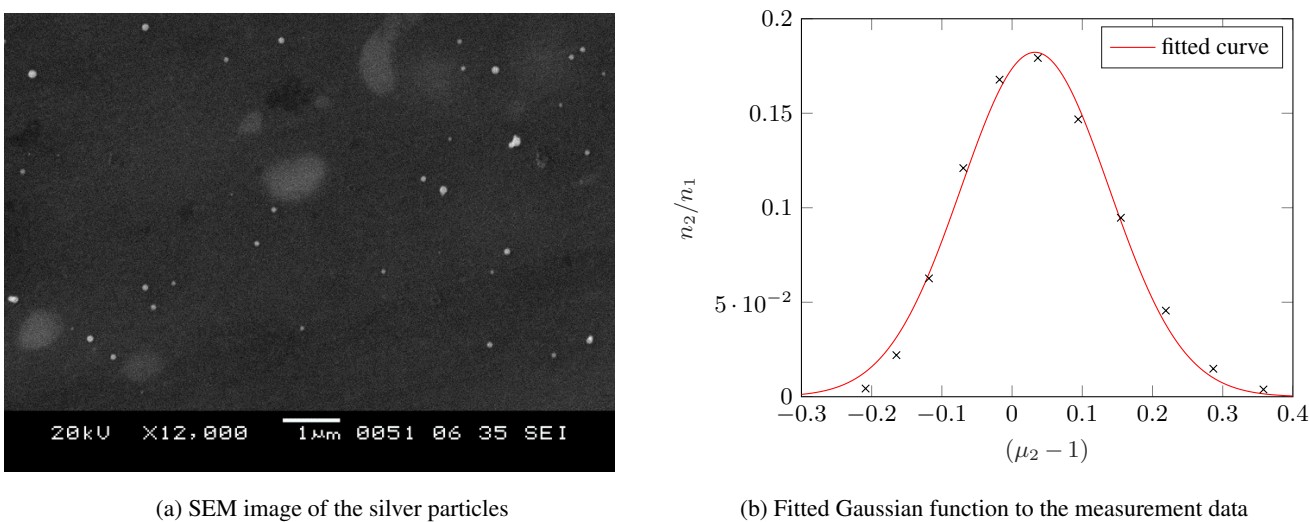

(a) SEM image of the silver particles

(b) Fitted Gaussian function to the measurement data

**Figure 8.** SEM image of silver particles a) and exemplary measurement data for measuring a transfer function at one operating point b)

Here $\Omega_1$ and $\Omega_2$ represent the transfer functions of the first and second measurement devices.

Deviating from the ideal type assumption of the previously discussed triangular transfer function, a Gaussian function is now assumed for the shape of the transfer function. This approach is sensible for considering the effects of diffusion or the lack of a plug-flow flow profile in the inlet gap. Moreover, the convolution of two Gaussian functions is also a Gaussian function, this





fits also very well to the measurement data (see Fig. 8b).

A Gaussian transfer function can be described by:

$$\Omega_1 = a \cdot \exp\left\{-\frac{(\widetilde{Z}-\widetilde{\mu}_1)^2}{c^2}\right\} \quad ; \quad \Omega_2 = d \cdot \exp\left\{-\frac{(\widetilde{Z}-\widetilde{\mu}_2)^2}{e^2}\right\} \tag{16}$$

Where $a$, $d$ are transfer function height fit parameters $\widetilde{\mu}_1$ and $\widetilde{\mu}_2$ are transfer function position fit parameters, and $c$, $e$ are transfer function width fit parameters. In the normalised case, the position fit parameters $\widetilde{\mu}_1$ and $\widetilde{\mu}_2$ should be 1, but for real

measurements, there are some minor errors or inaccuracies, that can be determined by these parameters. Since the position of the function on the abscissa is inconsequential when integrating from $-\infty$ to $+\infty$, the abscissa can be displaced arbitrarily. If the abscissa is now displaced so that the centre of the first transfer function is zero, the following equation is obtained:

$$n_2/n_1 = \frac{\int\limits_{-\infty}^{+\infty} a \cdot \exp\left\{-\frac{\widetilde{Z}^2}{c^2}\right\} \cdot d \cdot \exp\left\{-\frac{(\widetilde{Z}-\widetilde{\mu}_2+\widetilde{\mu}_1)^2}{e^2}\right\} \mathrm{d}\widetilde{Z}}{\int\limits_{-\infty}^{+\infty} a \cdot \exp\left\{-\frac{\widetilde{Z}^2}{c^2}\right\} \mathrm{d}\widetilde{Z}} \tag{17}$$

Solving these integrals:

$$\int\limits_{-\infty}^{+\infty} a \cdot \exp\left\{-\frac{\widetilde{Z}^2}{c^2}\right\} \cdot d \cdot \exp\left\{-\frac{(\widetilde{Z}-\widetilde{\mu}_2+\widetilde{\mu}_1)^2}{e^2}\right\} \mathrm{d}\widetilde{Z} = \sqrt{\frac{\pi}{c^2+e^2}} \cdot a \cdot d \cdot |c| \cdot |e| \cdot \exp\left\{-\frac{(\widetilde{\mu}_2-\widetilde{\mu}_1)^2}{c^2+e^2}\right\} \tag{18}$$

$$\int\limits_{-\infty}^{+\infty} a \cdot \exp\left\{-\frac{\widetilde{Z}^2}{c^2}\right\} \mathrm{d}\widetilde{Z} = \sqrt{\pi} \cdot a \cdot |c| \tag{19}$$

If the transfer function, i.e. $a$, $c$, of the first instrument is known, with the help of:

$$n_2/n_1 = \sqrt{\frac{1}{c^2+e^2}} \cdot d \cdot |e| \cdot \exp\left\{-\frac{(\widetilde{\mu}_2-\widetilde{\mu}_1)^2}{c^2+e^2}\right\} \tag{20}$$

it is possible to fit a Gaussian function to the measurement values and calculate the remaining parameters via comparison of the coefficients. Figure 8b presents the real measured values and the corresponding approximation of a Gaussian function.

The proposed method is easy to use and does not require a complex minimum search for all measurement points. In addition, the error caused by the influence of the start parameters for a minimum value search is eliminated. From equation (20), one can see that the height of the transfer function of the pre-classifying DMA is eliminated and therefore has no influence on the

measurement.

Since the pre-classifying instrument is a DMA, the determination of the transfer function of the CDMA with $U = 0\mathrm{V}$ must be adopted, so that the following equations can be applied:

$$\Omega_1 = a \cdot \exp\left\{-\frac{(\widetilde{\tau}-\widetilde{\mu}_1)^2}{c^2}\right\} \quad ; \quad \Omega_2 = d \cdot \exp\left\{-\frac{(\widetilde{\tau}-\widetilde{\mu}_2)^2}{e^2}\right\} \tag{21}$$

Therefore, the transfer function for the first DMA $\Omega_1(\widetilde{\tau})$ must be determined first. This is done in appendix C.





### 5.3 Measurement results

In general, three different particle sizes (50 nm, 100 nm, 200 nm) of spherical silver particles were analysed. Because the measured values were very similar for all particle sizes, only the results for a particle size of 100 nm are presented here. The other sizes are shown in appendix D. The transfer functions were determined over the full range of possible operating parameters. The aerosol flow rate was kept constant for each series of measurements, and the sheath air flow rate was varied so that the parameters could be determined for different values of $\beta$. Both the SMPS DMA and the CDMA were operated at the same ratio of aerosol to sheath air volume flow. The points on the red lines show comparative measurements of a DMA-DMA configuration as a reference. It should also be noted that the corrections explained in section 3.2 have already been applied here, so that the unwanted separation due to voltage or speed has already been eliminated.

To determine the transfer function of the pre-classifying DMA from the SMPS system, two identical DMAs were first measured in tandem configuration. Assuming that they are equal regarding their classification properties, it is possible to determine the relevant parameters for the pre-classifying DMA for DMA-measurements:

$$\mu_1 = 1.00319 \quad ; \quad c = 0.5868 \cdot \beta \tag{22}$$

and for AAC-measurements:

$$\mu_1 = 1.00319 \quad ; \quad c = 0.6468 \cdot \beta \tag{23}$$

### 5.3.1 DMA-Mode

Figure 9a plots the measured height of the transfer function against the aerosol to sheath air volume flow ratio. Comparing the measured values of the CDMA in DMA mode with the measurement values of a commercial DMA (TSI 3081), it is noticeable that the height is significantly lower. This is not surprising because the CDMA requires significantly more deflections and also a longer travel distance during which particles are deposited on the walls by impaction or diffusion, respectively. The scatter of the measured values can be explained by random variations in the operation of the CDMA. For example, at low aerosol volume flows and high values of $\beta$, only very small sheath air flows occur, which are outside the normal operating range of recirculation technology and do not guarantee a reliable and uniform volume flow. The very strong drop in the height of the transfer function from $\beta = 0.1$ to $\beta = 0.05$ is observed. There is also a slight drop in the height of the DMA transfer function, due to the increasing influence of diffusion as the transfer functions become narrower. However, this phenomenon does not explain the sharp decrease in the CDMA curves. In particular, for other particle sizes, exactly the same drop could be observed. This leads to the conclusion that there is no relevant particle loss due to diffusion or impaction, as a similar curve can be observed for all volume flows. It indicates that further particle losses are present, but the source could not be identified yet. To investigate this problem, a new prototype with a significantly shortened inlet and outlet regions of the aerosol and sample volume flow is required.

Figure 9b shows the standard deviation of the transfer function, which corresponds to the width of the transfer function. The green dashed line represents the ideal relationship between standard deviation and $\beta$. This can also be observed in the measured





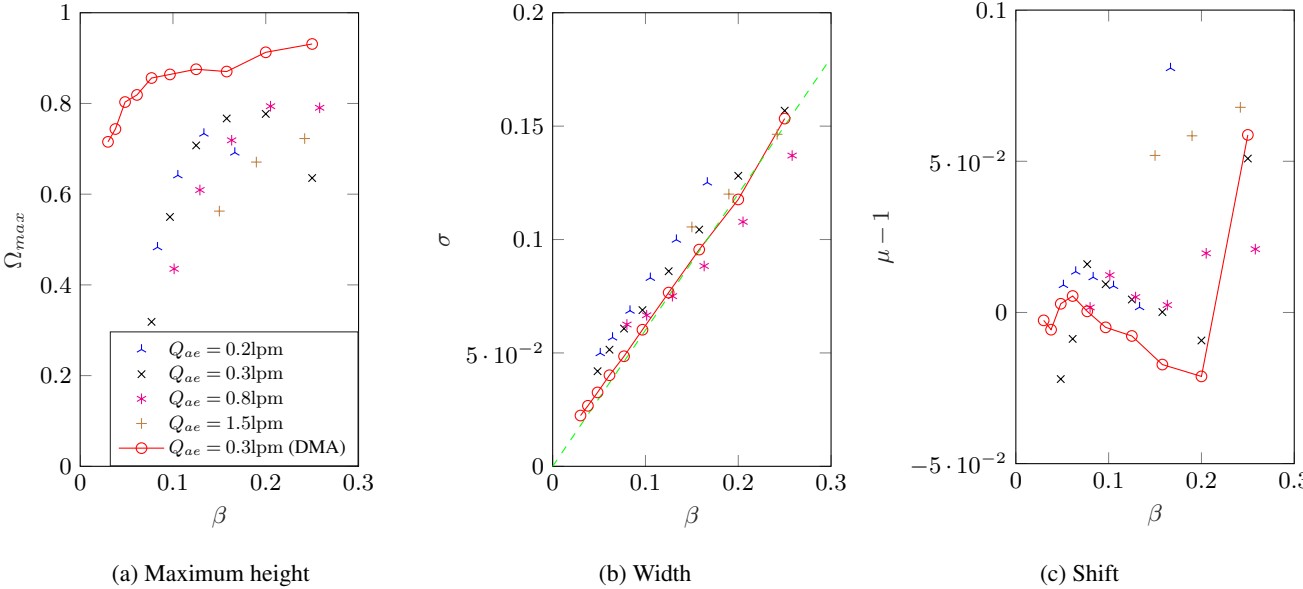

(a) Maximum height  (b) Width  (c) Shift

**Figure 9.** Measured transfer function parameters of the CDMA in DMA-mode for $d_p = 100$ nm

values, as they also exhibit a linear dependence on $\beta$. When the measured values are compared with the DMA reference values, it is noticeable that the width of the transfer function is generally slightly larger. This can be explained by the narrower gap of the CDMA at approximately the same length (the applied voltage is therefore significantly lower and therefore a lower Peclet number in the CDMA, hence diffusion plays a greater role) and also by the assembly and tolerances of the prototype (especially in combination with the short travelling distances of the particles in the radial direction).

Figure 9c shows the shift of the transfer function on the x-axis. It can be seen that there is no significant difference to the DMA comparison values. Larger deviations can be observed only for large $\beta$-values. This was again due to the control range of the sheath air flow.

### 5.3.2 AAC-Mode

It should be noted that the AAC measurements have already been corrected for charge distribution. This was performed by measuring the particle size distribution available directly at the outlet of the sintering furnace. As the charge distribution is known, it is possible to deduce the proportion of multiply charged particles that pass through the pre-classification. This fraction can then be subtracted from $n_1$ to obtain the actual number concentration of simply charged particles at the CDMA inlet.

Figure 10a shows the height of the transfer function for AAC mode. Unlike the DMA mode, there is no plateau from which the height remains constant. However, the transfer functions reach very high values. It can also be seen that the decrease is analogous to the heights shown for DMA. At low $\beta$ values, however, no measurement is possible, because at $\beta < 0.1$ almost



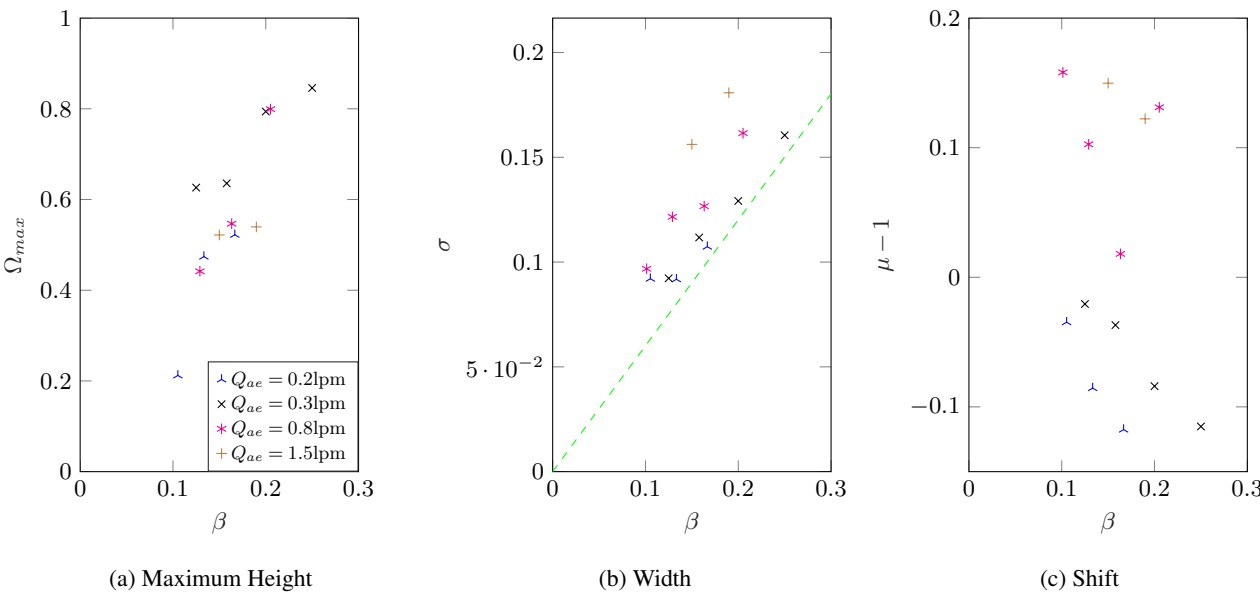

(a) Maximum Height          (b) Width          (c) Shift

**Figure 10.** Measured transfer function parameters of the CDMA in AAC-mode for $d_p = 100$ nm

all particles are separated by the centrifugal force at the inlet and outlet of the CDMA.

Figure 10b again shows the standard deviation of the transfer function. There is still a linear dependence on the $\beta$ values and fluctuation around the ideal value. However, the scattering of the measured values was significantly larger compared to the DMA-mode. This is due to the more difficult flow control as the flows take a longer distance to fully developed, i.e. due to the rotation the walls have to take all the air with in the rotation, too. By changing for example the sheath air, CFD simulations have shown that the development of the flow profile is delayed in the axial direction. This effect becomes stronger as the sheath

air flow rate increases. One potential improvement is the installation of vanes at the point at which the sheath radius changes. Moreover, higher angular velocities lead to a higher pressure difference at the inlet, and hence, some flow disturbances may occur there. This can affect the classification zone, and explains larger variations in the results. In addition, the strong separation at the inlet and outlet area results in very high correction factors, especially for small beta values.

Figure 10c shows a similar picture as Fig. 10b. Again, the deviation is significantly greater than that in DMA-mode, which,

as in Fig. 10b, is due to the flow field in CDMA. However, except one outlier, in Fig. 10c seems to be a linear dependence on the beta values for each aerosol volume flow. In addition, the curves were all parallel, with the curves shifting in the positive direction as the aerosol volume flow increased. Again, this can be explained by the difficulty of flow adaptation at higher speeds, i.e. higher flow velocities (or volume flows). The deviation might also be caused by manufacturing tolerances or imperfectly aligned system components. As the deviation for the other particle sizes is also very similar to this results, a calibration for the

CDMA could be performed during further validation.



## 6 Conclusions

As described in the previous section, the transfer functions of the CDMA for $\widetilde{\tau} = 0$ and $\widetilde{Z} = 0$ agreed well with the theory and reference values of a DMA-DMA configuration. For $\beta < 0.2$ the height of the transfer function decreased much more than for $\beta > 0.2$ but the cause has not yet been conclusively clarified. In order to find the cause, a new prototype is to be designed and built that significantly reduces the depositions in the inlet and outlet areas and minimises the electric fields outside the classification zone. The superposition of these effects makes it difficult to determine the cause in the current arrangement. As one operating point is sufficient for the initial investigations, further investigations will be conducted for $\beta = 0.2$. For this purpose, this operating point needs to be measured again in detail and possibly a calibration of the CDMA needs to be carried out. Furthermore, a validation for particle sizes larger than 200 nm is necessary in order to be able to fully describe the CDMA for those larger particles as well. The ideal 2-dimensional transfer function can be calculated using the particle trajectory method. However, an extension to the streamline model is highly recommended for validation, because diffusion in the classifier gap can be considered. Furthermore, a new robust method of measuring transfer functions was presented, which made it possible to draw conclusions about the transfer function simply by measuring the number concentrations. If these results agree with the theoretical values for different particle types, the final step is to develop an algorithm that can be used to calculate a 2D-distribution with respect to $d_m$ and $d_{st}$ from the measured values.

## Appendix A: Comment for the definition of the particle relaxation time

Contrary to *Tavakoli and Olfert*, the Stokes equivalent diameter was used instead of the aerodynamic equivalent diameter. Furthermore, the argument of the Cunningham slip correction should be mentioned here. The equations for calculating the slip correction are derived only for spherical particles. There are many studies in the literature dealing with the slip correction for non-spherical particles. For example, *Cheng et al.* and *Dahneke* propose the introduction of a new equivalent diameter for non-spherical particles with the same slip correction. However, these methods are mostly limited to specific particle shapes and apply only to the free molecular or continuum region but not to the transition region. However, we consider the aerodynamic equivalent diameter not to be appropriate to calculate the slip correction as done by *Tavakoli and Olfert*, since the aerodynamic diameter depends on particle density as well. Namely, two particles of identical shape but different densities would experience the same drag force and thus slip correction, while having different aerodynamic diameters. Therefore, we suggest to use the mobility equivalent diameter instead. This is in accordance with a number of studies in the literature, such as (Knutson and Whitby, 1975; Sorensen, 2011). Moreover, slip correction experimental investigations have typically been performed using the Millikan apparatus, where the particles are moved in an electric field (Buckley and Loyalka, 1989; Rader, 1990; Allen and Raabe, 1985).



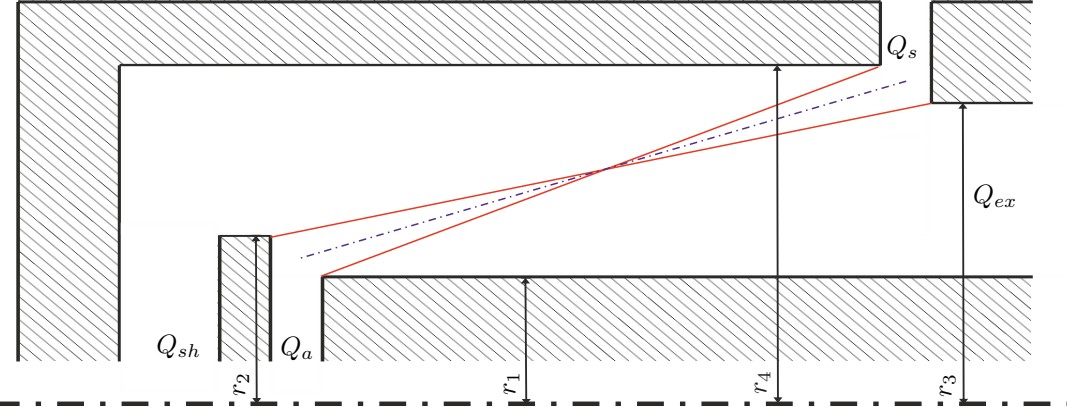

**Figure B1.** Characteristic Particle Paths: minimal and maximal particle paths (solid red lines) and centred particle path (dash-dotted blue line) for combinations of $\tau$ and $Z$

.

### Appendix B: Derivation of the two-dimensional transfer function using the particle trajectory calculation

In accordance with the specified operating parameters (i.e. voltage $U$ and speed $\omega$), a multitude of combinations of the equivalent particle sizes $d_m$ and $d_{st}$, or combinations of $\tau$ and $Z$, can be derived analytically from the geometry and the volume flow conditions. The aforementioned combinations are subject to a probability of classification. A characteristic combination, for instance, exhibits the property that if a particle is introduced at the centre of the inlet gap, it will also be collected at the centre of the classifying gap. In the case of edge transfer functions, the following relationships are observed: $\tau = \tau^*$, or $\widetilde{\tau} = 1$ (for $Z^* = 0$), and $Z = Z^*$, or $\widetilde{Z} = 1$ (for $\tau^* = 0$). The results are presented in section 4.2 of this article.

Two characteristic classification properties are the limiting particle trajectories, further on called maximum and minimum particle path. All particles that traverse the classifying section at a shorter distance than the particle at the minimum path are no longer classified. This minimum path is defined by substituting $r_{in} = r_2$ and $r(L) = r_3$ into equation (8). In contrast, the maximum particle trajectory represents the boundary from which particles traverse a greater distance are deposited at the inner electrode ($r_{in} = r_1$ and $r(L) = r_4$). Equation (8) may be employed to ascertain the transfer probability for each tau-Z combination. If two critical radii ($r_c$) are defined, where $r_1 < r_c < r_{c2} < r_2$ and a constant particle flux density at the inlet is assumed, the transfer probability can be determined by the following equation:

$$\Omega_{CDMA} = \max\left[\min\left(f_1, f_2, 1\right), 0\right] \tag{B1}$$

$$f_1 = \frac{r_2^2 - r_c^2}{r_2^2 - r_1^2} - \max\left(\frac{r_2^2 - r_{c2}^2}{r_2^2 - r_1^2}, 0\right) \tag{B2}$$





$$f_2 = \frac{r_{c2}^2 - r_1^2}{r_2^2 - r_1^2} - \max\left(\frac{r_c^2 - r_1^2}{r_2^2 - r_1^2}, 0\right) \tag{B3}$$

In other words, if we assume that the minimum particle path is limiting (as is the case for $f_1$ with the DMA transfer function

(Wang and Flagan, 1990)), a lower critical radius can be defined so that all particles entering at $r < r_c$ are no longer classified, but are separated behind the classifying gap or discharged with the sheath air.

Therefore $r_c$ is defined by:

$$r_3 = \sqrt{r_c^2 \cdot \exp\left\{\widetilde{\tau} \cdot \frac{2\widetilde{h}}{1+\beta}\right\} + \bar{r}^2 \cdot \frac{\widetilde{Z}}{\widetilde{\tau}} \cdot \left[\exp\left\{\widetilde{\tau} \cdot \frac{2\widetilde{h}}{1+\beta}\right\} - 1\right]} \tag{B4}$$

As a consequence of the speed-controlled operating mode, it is possible that not all particles entering at $r_2$ will be classified.

Because of the elevated feed radius, a higher drift velocity is observed at the outlet of the transfer section, thereby enabling the separation of particles at the outer electrode or excess air (for further details, please refer to section 4.2). In order to accommodate this phenomenon, a second critical radius, designated as $r_{c2}$, is introduced, situated between $r_c$ and $r_2$; the calculation is performed in accordance with the following formula:

$$r_4 = \sqrt{r_{c2}^2 \cdot \exp\left\{\widetilde{\tau} \cdot \frac{2\widetilde{h}}{1+\beta}\right\} + \bar{r}^2 \cdot \frac{\widetilde{Z}}{\widetilde{\tau}} \cdot \left[\exp\left\{\widetilde{\tau} \cdot \frac{2\widetilde{h}}{1+\beta}\right\} - 1\right]} \tag{B5}$$

Once the flow ratio $\beta$ has been established, it can then be subtracted from the initial value in order to prevent the value $r_{c2}$ from exceeding $r_2$. The minimum value for this term is set to 0.

In light of the continuously increasing probability of transfer in the direction of $r_1$, a new parameter $f_2$ is introduced. It is important to note that the maximum particle path represents a limiting quantity, too. The same procedure is applied as for $f_1$, but now $r_{c2}$ is used as the critical radius for the first term, while $r_c$ is used in the second term. This implies that all particles

entering the transfer section with a radius greater than $r_{c2}$ are separated at the outer electrode before the classification gap. As a consequence of the reduction in radius, particles entering the transfer section at $r_{in} > r_1$ are again subjected to separation. A second critical radius is defined at which the particles are classified. The area ratio can once more be subtracted from the first term. Since the case $r_c < r_1$ cannot occur, a value of zero is also defined here as the minimum.

The probability of a $\tau - Z$ combination being classified is now the smaller of $f_1$ and $f_2$. This is due to the fact that both the maximum and the minimum particle path were taken into account. Furthermore, it should be noted that the transfer function can assume a maximum value of 1. It is also necessary to ensure that the transfer function does not become smaller than 0. This implies that the minimum of $f_1$, $f_2$ and 1 must not become smaller than 0 (cf. equation 10).

Upon substituting equations (B4) and (B5) into equations (B2) and (B3), the following result is obtained:

$$f_1 = \frac{r_2^2 - (r_3^2 + \bar{r}^2 \cdot \frac{\widetilde{Z}}{\widetilde{\tau}}) \cdot \exp\left\{-\widetilde{\tau} \cdot \frac{2\widetilde{h}}{1+\beta}\right\} + \bar{r}^2 \cdot \frac{\widetilde{Z}}{\widetilde{\tau}}}{r_2^2 - r_1^2} - \max\left(\frac{r_2^2 - (r_4^2 + \bar{r}^2 \cdot \frac{\widetilde{Z}}{\widetilde{\tau}}) \cdot \exp\left\{-\widetilde{\tau} \cdot \frac{2\widetilde{h}}{1+\beta}\right\} + \bar{r}^2 \cdot \frac{\widetilde{Z}}{\widetilde{\tau}}}{r_2^2 - r_1^2}, 0\right) \tag{B6}$$





$$f_2 = \frac{(r_4^2 + \bar{r}^2 \cdot \frac{\widetilde{Z}}{\widetilde{\tau}}) \cdot \exp\left\{-\widetilde{\tau} \cdot \frac{2\widetilde{h}}{1+\beta}\right\} - \bar{r}^2 \cdot \frac{\widetilde{Z}}{\widetilde{\tau}} - r_1^2}{r_2^2 - r_1^2} - \max\left(\frac{(r_3^2 + \bar{r}^2 \cdot \frac{\widetilde{Z}}{\widetilde{\tau}}) \cdot \exp\left\{-\widetilde{\tau} \cdot \frac{2\widetilde{h}}{1+\beta}\right\} - \bar{r}^2 \cdot \frac{\widetilde{Z}}{\widetilde{\tau}} - r_1^2}{r_2^2 - r_1^2}, 0\right) \tag{B7}$$

Assuming constant flux density at the inlet, the radii can be set in relation to each other. This yields the following relationships:

$$u = \frac{Q_{sh} + Q_a}{\pi(r_4^2 - r_1^2)} = \frac{Q_{sh}}{\pi(r_4^2 - r_2^2)} = \frac{Q_a}{\pi(r_2^2 - r_1^2)} = \frac{Q_{ex}}{\pi(r_3^2 - r_1^2)} = \frac{Q_s}{\pi(r_4^2 - r_3^2)} \tag{B8}$$

Using these relations for the typical operation condition ($Q_a = Q_s$ ans subsequently $Q_{sh} = Q_{ex}$) and $\beta = Q_a/Q_{sh}$ can lead to:

$$\left(\frac{r_2}{r_4}\right)^2 = \frac{\beta + \kappa^2}{1+\beta} \quad ; \quad \left(\frac{r_3}{r_4}\right)^2 = \frac{1/\beta + \kappa^2}{1/\beta + 1} \quad ; \quad \left(\frac{\bar{r}}{r_4}\right)^2 = \frac{(\kappa+1)^2}{4} \tag{B9}$$

With:

$$\kappa = \frac{r_1}{r_4} = \frac{1 - \widetilde{h}/2}{1 + \widetilde{h}/2} \tag{B10}$$

In order to insert the ratios from equations 9 and 10, it is necessary to expand the denominator and numerator with $r_4^2$ in equations 7 and 8. From this follows for $f_1$ and $f_2$:

$$f_1 = \frac{\frac{\beta + \kappa^2}{1+\beta} - \left(\frac{1/\beta + \kappa^2}{1/\beta + 1} + \frac{(\kappa+1)^2}{4} \cdot \frac{\widetilde{Z}}{\widetilde{\tau}}\right) \cdot \exp\left\{-\widetilde{\tau} \cdot \frac{2\widetilde{h}}{1+\beta}\right\} + \frac{(\kappa+1)^2}{4} \cdot \frac{\widetilde{Z}}{\widetilde{\tau}}}{\frac{\beta + \kappa^2}{1+\beta} - \kappa^2} - ...$$

$$...\max\left(\frac{\frac{\beta + \kappa^2}{1+\beta} - \left(1 + \frac{(\kappa+1)^2}{4} \cdot \frac{\widetilde{Z}}{\widetilde{\tau}}\right) \cdot \exp\left\{-\widetilde{\tau} \cdot \frac{2\widetilde{h}}{1+\beta}\right\} + \frac{(\kappa+1)^2}{4} \cdot \frac{\widetilde{Z}}{\widetilde{\tau}}}{\frac{\beta + \kappa^2}{1+\beta} - \kappa^2}, 0\right) \tag{B11}$$

$$f_2 = \frac{\left(1 + \frac{(\kappa+1)^2}{4} \cdot \frac{\widetilde{Z}}{\widetilde{\tau}}\right) \cdot \exp\left\{-\widetilde{\tau} \cdot \frac{2\widetilde{h}}{1+\beta}\right\} - \frac{(\kappa+1)^2}{4} \cdot \frac{\widetilde{Z}}{\widetilde{\tau}} - \kappa^2}{\frac{\beta + \kappa^2}{1+\beta} - \kappa^2} - ...$$

$$...\max\left(\frac{\left(\frac{1/\beta + \kappa^2}{1/\beta + 1} + \frac{(\kappa+1)^2}{4} \cdot \frac{\widetilde{Z}}{\widetilde{\tau}}\right) \cdot \exp\left\{-\widetilde{\tau} \cdot \frac{2\widetilde{h}}{1+\beta}\right\} - \frac{(\kappa+1)^2}{4} \cdot \frac{\widetilde{Z}}{\widetilde{\tau}} - \kappa^2}{\frac{\beta + \kappa^2}{1+\beta} - \kappa^2}, 0\right) \tag{B12}$$

## Appendix C: Calculation of the transfer functions of the pre-classifying DMA

To determine the transfer function of the pre-classifying DMA, two identical DMAs are measured first. It is assumed that the transfer functions are identical. Which means:

$$d = a \quad ; \quad e = c \tag{C1}$$

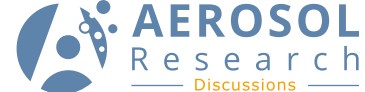

Also assuming that the shift $\widetilde{\mu}$ of the DMA's has the same distance to $\widetilde{\mu}_1$ and $\widetilde{\mu}_2$, the shift of the pre-classifying DMA can be calculated $\widetilde{\mu} = (\widetilde{\mu}_2 - \widetilde{\mu}_1)/2$. In the DMA mode (i.e. omega = 0), equation (20) could be simplified:

$$n_2/n_1 = \sqrt{\frac{1}{2}} \cdot d \cdot \exp\left\{-\frac{(\widetilde{\mu}_2 - \widetilde{\mu}_1)^2}{2 \cdot e^2}\right\} \tag{C2}$$

The parameters now can be extracted by fitting a Gaussian function to the measurement data and compare the coefficients of the fitted curve and equation (C2).

As already explained in section 5.2, a validation of the AAC mode (i.e. $\omega = 0$) is only possible if the transfer function of the DMA can be expressed as a function of the particle relaxation time. For the measurement results $n_2/n_1$, the mobility must be converted into the particle relaxation time assuming particle shape. A renewed application of a Gaussian function enables the parameters for $\Omega(\widetilde{\tau})$ to be determined in the same way as for the DMA mode.

An alternative approach is to plot $\Omega(\widetilde{Z})$, convert the mobilities into particle relaxation times and approximate a Gaussian

function by the values in the Omega-tau diagram. The resulting approximation is shown in Fig. C1. As illustrated in the Fig., there are inconsistencies between the fitting function and the generated values, particularly in the marginal areas, since the measured distribution appears to be slightly skewe. Nevertheless, the shape is relatively similar, allowing for an approximation of the curve. With the assistance of the parameters from section 5.3, the width of the curve is determined to be $\sigma_{U=0} = 0.06468$ ($\sigma_{\omega=0} = 0.05868$). However, for an exact calculation, the aforementioned method should be employed.

These approaches for the determination of $\Omega(\widetilde{\tau}))$ are only valid for spherical particles!

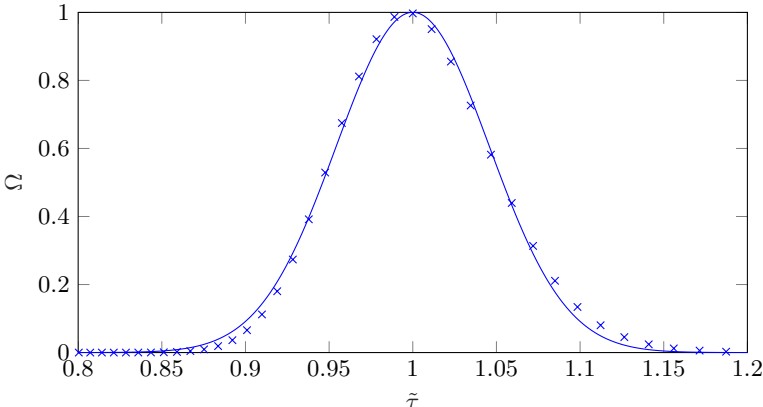

**Figure C1.** Fit of a Gaussian function to the $\Omega - \widetilde{\tau}$ domain, based on the mobility measurements




## Appendix D: Measurement values for the determination of the pre-classifying DMA parameters

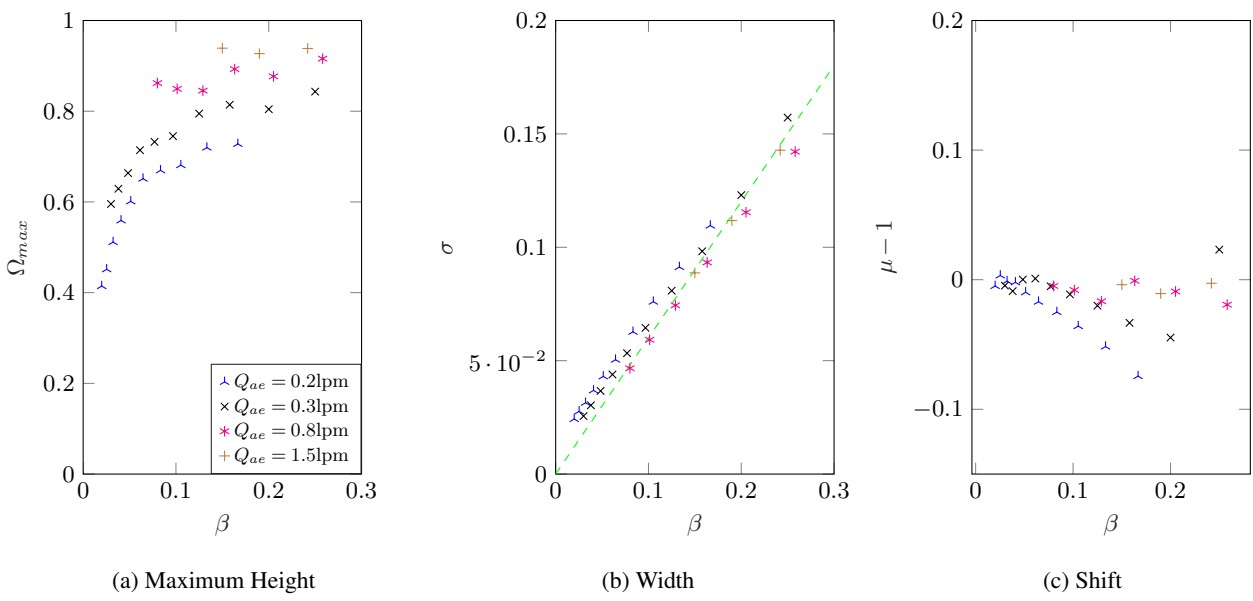

**Figure D1.** Measured transfer function parameters of the pre-classifying DMA for $d_p = 50$ nm

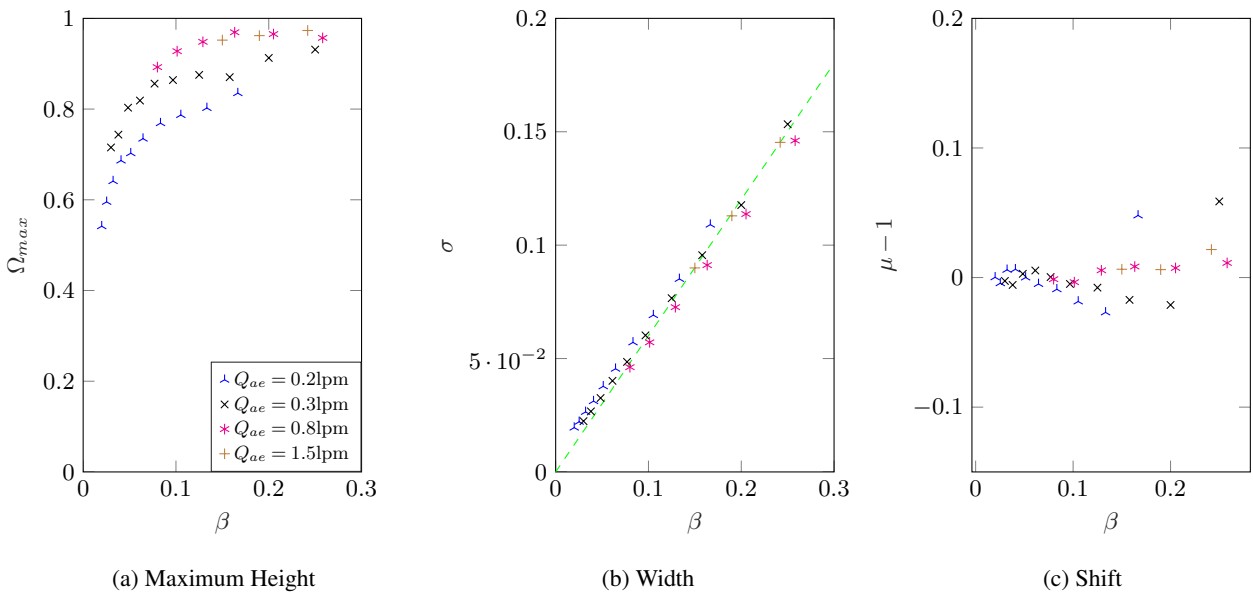

**Figure D2.** Measured transfer function parameters of the pre-classifying DMA for $d_p = 100$ nm



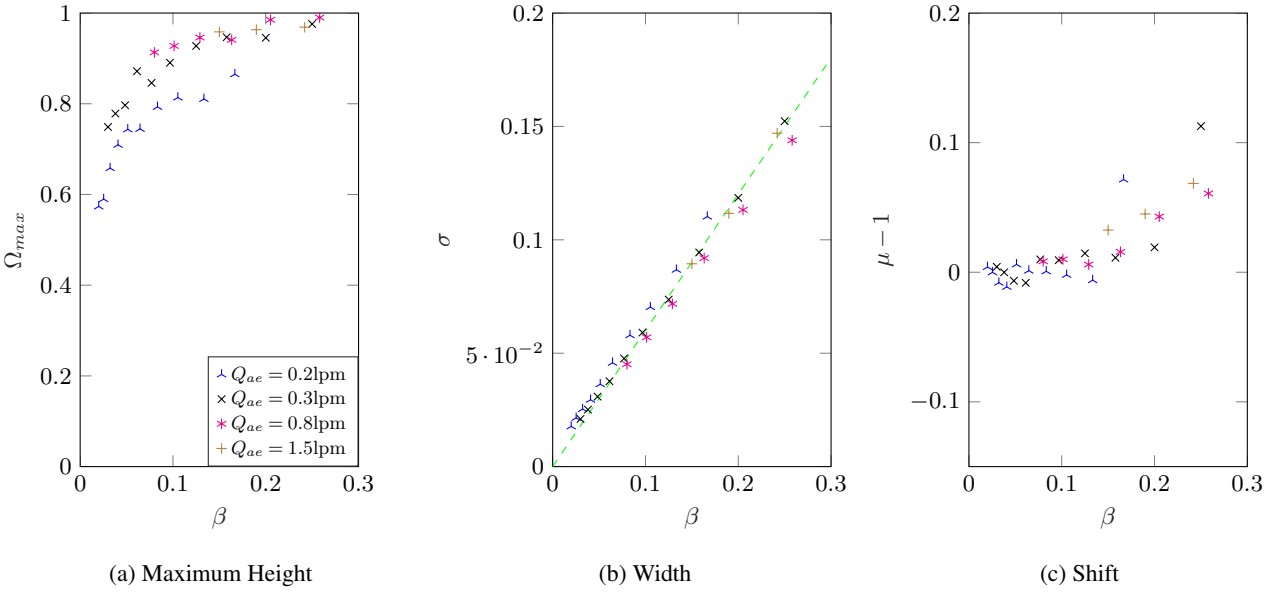

(a) Maximum Height         (b) Width         (c) Shift

**Figure D3.** Measured transfer function parameters of the pre-classifying DMA for $d_p = 200$ nm





## Appendix E:  Measurement Values for 50nm and 200nm

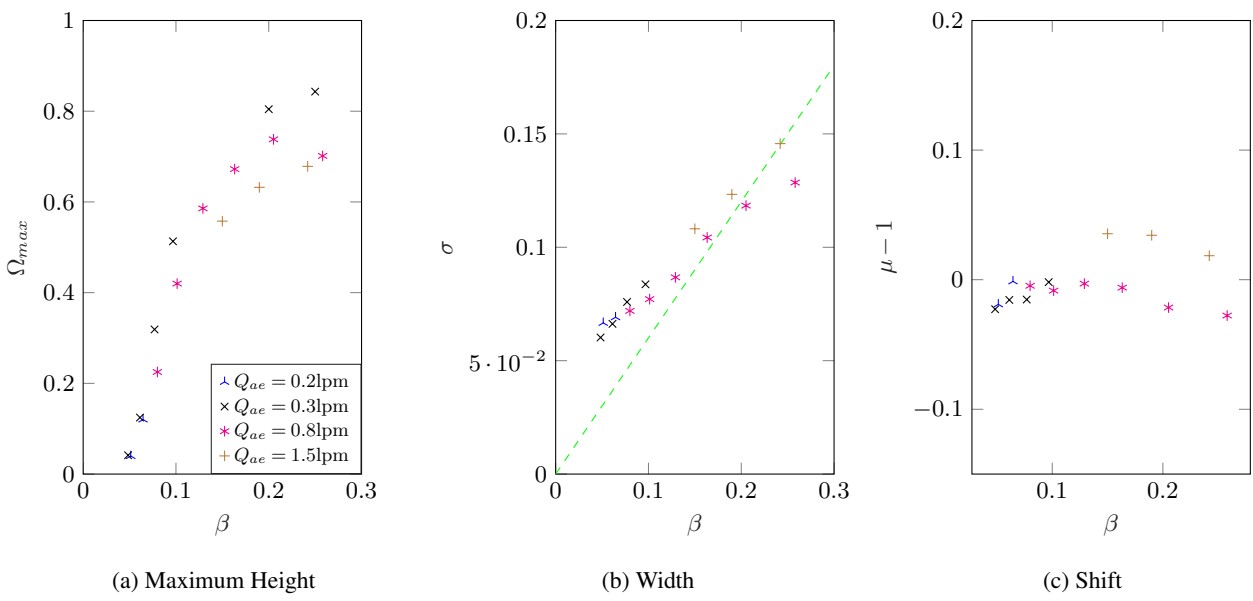

**Figure E1.** Measured transfer function parameters of the CDMA in DMA-Mode for $d_p = 50$ nm

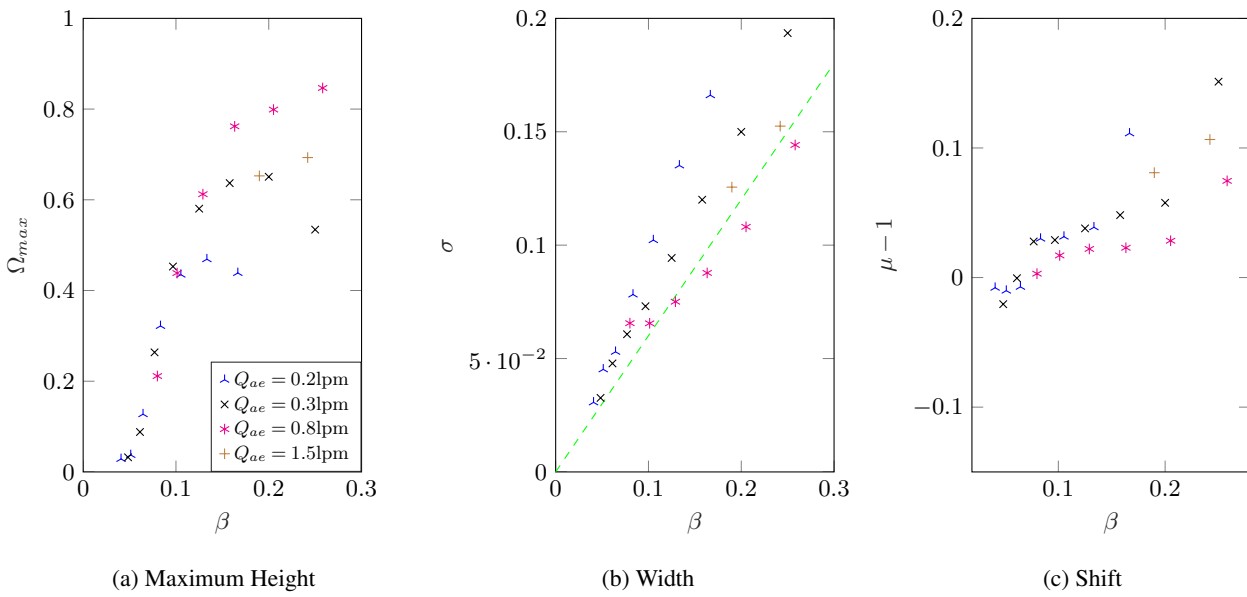

**Figure E2.** Measured transfer function parameters of the CDMA in DMA-Mode for $d_p = 200$ nm





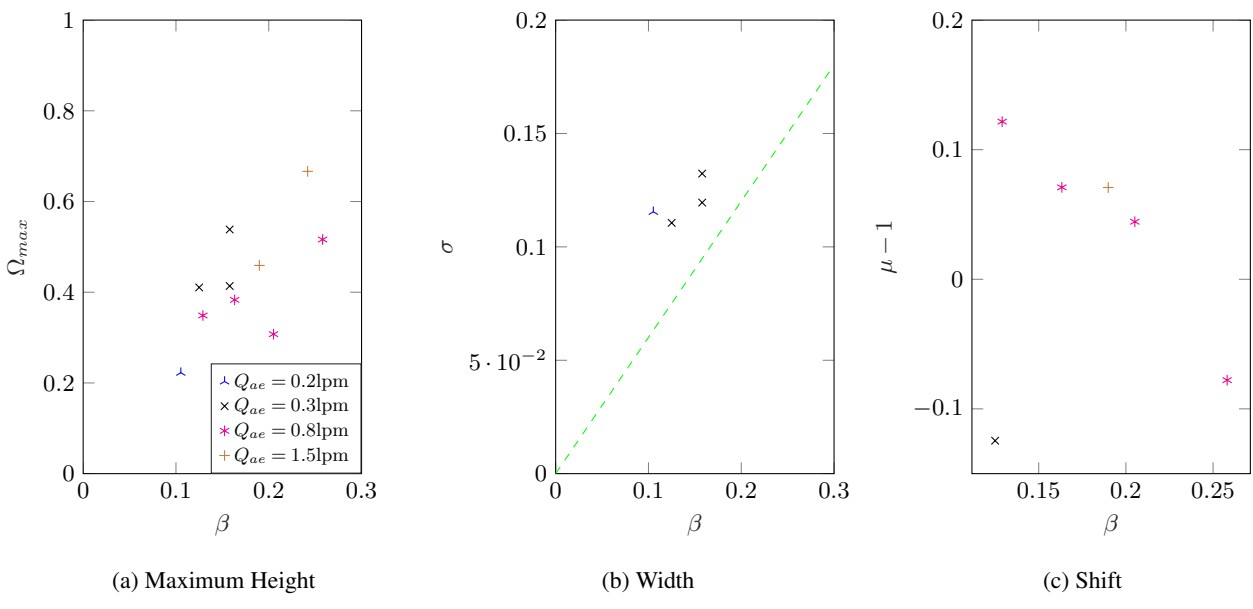

(a) Maximum Height      (b) Width      (c) Shift

**Figure E3.** Measured transfer function parameters of the CDMA in AAC-Mode for $d_p = 50$ nm

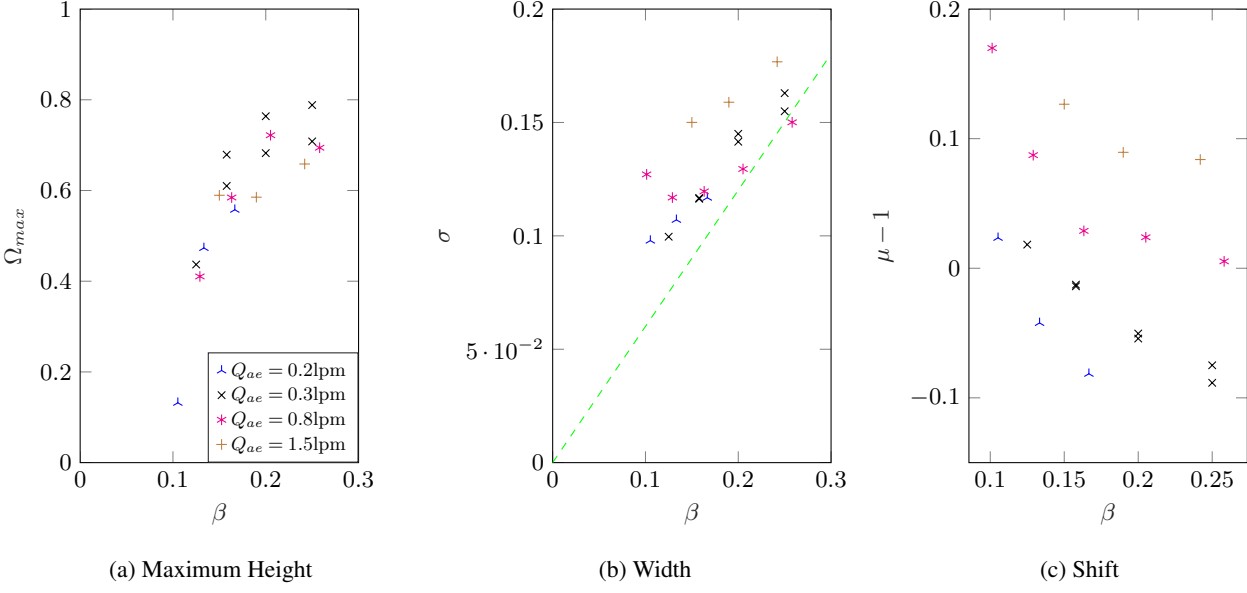

(a) Maximum Height      (b) Width      (c) Shift

**Figure E4.** Measured transfer function parameters of the CDMA in AAC-Mode for $d_p = 200$ nm

*Author contributions.* **Torben Rüther:** Software, Validation, Formal analysis, Investigation, Data Curation, Writing - Original Draft, Visualization **David Rasche:** Conceptualization, Methodology, Formal analysis, Writing - Review and Editing, Funding acquisition **Hans-Joachim**
**Schmid:** Resources, Writing - Review and Editing, Supervision, Project administration, Funding acquisition



*Competing interests.* The authors declare that they have no known competing financial interests or personal relationships that could have appeared to influence the work reported in this paper.





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
