# Peer review of "The Centrifugal Differential Mobility Analyser - Concept and initial validation of a new device for measuring 2D property distributions"

_Aerosol Research, 2024_

## Author Comment (AC1)

**We gratefully acknowledge the effort of the referee in thoroughly evaluating the manuscript. He provided very valuable comments which helped to improve the manuscript significantly.**

**Major concerns:**

1.  The use of the different equivalent diameters is confusing. In the abstract, it is mentioned that the CDMA could measure size distributions based on the mobility and Stokes equivalent diameter. To the best of my knowledge, these two are the same. Apparently, this should be the mobility (or Stokes) and aerodynamic diameter. After reading Appendix A, I can follow the authors' arguments for using the mobility diameter in the calculation of the Cunningham factor and thus the particle relaxation time. For a spherical particle, the mobility (or Stokes) diameter equals the geometric diameter, which describes the curvature of the sphere's surface, which is responsible for the molecular slip. Nonetheless, the classification in the AAC mode is still density-dependent and consequently based on the aerodynamic and not the Stokes diameter.

    Indeed there are different definitions of the Stokes diameter in literature. We regret that our text was not clear on this point. We tried to give clear definitions, now to avoid any misunderstanding.

    In most publications in this field, the mobility equivalent diameter is the diameter or the sphere, which experiences the same drag force for a given relative velocity. In case of an electric mobility analysis, the classification velocity w is given by a balance of electrical and drag force. If the charge Q and the electric field E is assumed to be known the mobility diameter is directly related to the classification velocity w:

    $$d_m = Q \cdot E \cdot Cc/(3 \cdot pi \cdot \eta \cdot w)$$

    In case of settling in any field force (e.g. gravitation, centrifugation), the classification velocity is given by a balance of the field force (i.e. proportional to mass) and the drag force:

    $$F_{field} = F_{Drag}$$
    $$m \cdot b = 3 \cdot pi \cdot \eta \cdot w \cdot d_m/Cc$$
    $$\rho \cdot \pi/6 \cdot d_v^3 \cdot b = 3 \cdot \pi \cdot \eta \cdot w \cdot d_m/Cc$$

    Here, $m$ is the particle mass and $b$ is the acceleration. For the settling velocity under gravity b = g. In this case, a centrifugal force is applied, so that $b = \omega^2 \cdot r$.

    $$d_{st}{}^2 = d_v^3/d_m = 18 \cdot \eta \cdot w/(\rho \cdot \omega^2 \cdot r \cdot Cc)$$

    $d_m$ is the mobility equivalent diameter, $d_v$ the volume equivalent diameter and $d_{st}$ the stokes equivalent diameter.

    Therefore, the Stokes diameter is not equal to the mobility diameter. The Stokes diameter is similar to the aerodynamic diameter, where the density is the material density and not set unity.

    $$d_{st} = d_{ae} \cdot \sqrt{\rho_0/\rho}$$
* * *
**2  Concept and fundamental theory**

The newly developed principle of CDMA combines the concepts of DMA and AAC. In general, the CDMA consists of two concentric cylinders between which high voltage can be applied, and/or both of which can be rotated at the same angular speed. This means that both the voltage and the speed can be superimposed, whereas in the DMA, only the voltage can be varied, and in the AAC, only the speed. This means that with the CDMA, particles can be classified according to their mobility equivalent diameter (i.e. a spherical particle of this diameter experiences the same drag force for a given relative velocity as the actual particle) and Stokes equivalent diameter (i.e. that is the particle size of a spherical particle which has the same settling velocity, while having the same density (Colbeck, 2013; Reist, 1993), therefore depending on drag force and mass)[1]. By measuring all
* * *
[1]Often the aerodynamic diameter is used instead of the stokes diameter. Whereas the aerodynamic diameter standardizes the shape and density of the particle, the Stokes diameter standardizes the shape only (Baron et al., 2011). Since, in this paper we focus on particle shape, the Stokes diameter is used. However the Stokes diameter could be easily replaced by the aerodynamic diameter.

2. From the title of the manuscript, I would have expected a fully characterized device. However, the conclusion is that improvements are needed and shall be implemented in a second prototype, for which a more comprehensive evaluation will be conducted. I therefore suggest to make this clearer in the title by including terms like "concept" or "initial validation".

Thank's for that comment. we changed the title to:

The Centrifugal Differential Mobility Analyser - Concept and initial validation of a new device for measuring 2D property distributions

3. The manuscript uses many equations with a multitude of different symbols. I found it a bit cumbersome to search for the meaning of the different symbols, at places where they were not used for the first time. I therefore suggest to add a nomenclature in the beginning or in an appendix (depending on the publisher's policy on that).

This is also a good note. We've added a nomenclature in the manuscript, and we'll add it as a supplement in this discussion.

4. It seems that all trajectory calculations assumed plug flow and no hyperbolic flow. Is this justified or a simplification? This should be discussed in the manuscript.

It's a good point. So, we added a comment on that.
* * *
**4.1 Two-dimensional transfer function based on the particle trajectory calculation**

Since in CDMA, as in DMA and AAC, the ratio of aerosol volume flow to sheath air volume flow cannot be infinitely small and the inlet and sample gaps are also finite, the classified aerosol is not completely monomodal, but a distribution exists. Assuming a constant particle flux density at the inlet, stratified plug flow, and homogeneous E-field in the classifying gap, as well as non-inertial and diffusion-free particles, this distribution can be calculated analytically. The assumption of a plug flow profile is justified, since for a particle which is classified mainly the mean velocity is relevant, because the particle path crosses the whole flow domain. Stolzenburg (1988) proved this with the derivation of transfer functions based on a streamline approach for diffusing particles. The derivation of the two-dimensional transfer function is given in Appendix B.
* * *
In addition, we investigated this in more detail, which is however out of scope of this paper. So, on the one hand we also derived the transfer functions based on streamline function approach like Stolzenburg did (Topic of my next publication which is also already submitted). And for that, only minor deviations can be seen between a laminar flow profile and the plug flow velocity profile. Moreover, we did a CFD calculation for different rotational speeds and used this velocity data to calculate the transfer functions again (this will be in a further publication). There you can see that for no rotation the transfer function matches the ideal values, well. For some rotation, the velocity profile is stratified, but not completely laminar and the profile also changes within the classification zone. This leads to a distortion of the transfer function.

[Figure]

(a) Pure electric mode, at 20.99 V.

(b) Pure rotational mode, at 475 RPM and $\rho = 10490\,\text{kg/m}^3$.

(c) Pure rotational mode, at 2000 RPM and $\rho = 580\,\text{kg/m}^3$.

Figure 13. Simulated transfer functions, for a mean particle size of 100 nm for virtual particles, for pure electric and pure rotation modes, in comparison to the ideal transfer functions derived by streamline theory.

**Minor Concerns:**

Line 13: The first sentence of the introduction does not really have any content. I suggest eliminating it.
Done.

Line 15: A DMA classifies particles based on the (electrical) mobility diameter. I only know the term "hydrodynamic diameter" from the characterization of particles in liquids.
The hydrodynamic diameter is also mostly used for the drag force in liquds, so it's quite like the mobility diameter. But you are right, for this it should be the mobility diameter. We changed it in the document.

Lines 18ff: The discussion on particle surface area appears completely out of the blue and is not picked up again in the manuscript. Either explain in more detail why this is important for this work or eliminate this discussion.
We agree and replaced "surface area" by "shape".

Line 29: What is described here as tandem setups does not necessarily contain two or more measurement systems, but usually just different classification systems.
We replaced "measurement" with "classification"

Line 51: The flow is not applied to the inner cylinder, but introduced near the cylinder.
Done.

Line 56/equation 1: The force balance is typically written as $q \cdot E + F_{drag} = m \cdot a$. Here the drag force will receive a negative sign due to the direction of the relative velocity between particle and flow. In the way that equation (1) is set up, the signs do not seem to fit.
We think it depends in which direction the electric field is defined. So, if it is defined as in figure 1, the electric force on positively charged particles will be in the same direction as the centrifugal force. Thus, equation (1) should be correct, in this case. To make it clear, we added a reference to the figure.

Line 65: it should read inner $r_1$ and outer $r_4$ radius.
Right, done.

Lines 91ff: Please provide information on the flow rates of the CDMA.
Done.

Line 92: The density should be either 1 g/cm³ or 1000 kg/m³
Right, done.

Line 94: I don't understand this. Ok, you need a suitable sealing for the device, but why "for both the mobility and the Stokes diameter" (again, Stokes should be aerodynamic diameter, but that is not the reason why I don't understand the sentence).
Again you are right, this isn't depending on the diameters. It's a problem for devices, where a relative velocity between different parts exists. We deleted the first part of the sentence.

Lines 98/99: It would be helpful if you could indicate the location of these seals in Figure 2.
Done.

Lines 99-109: 1) Which polarity do you apply, i.e. which particle polarity do you classify?; 2) A voltage is always applied between two electrodes by applying different potentials to the individual electrodes, 3) Why is the high potential applied to the outer cylinder and the low potential (ground) to the inner cylinder? What about precautions for user safety?

1) With the CDMA, due to the centrifugal forces it is possible to classify both polarities. But surely for one polarity the acting force is in the direction of the centrifugal force. Since we apply a negative Voltage, the positively charged particles will go to the outer cylinder, negatively charged particles will only be classified in case of centrifugal forces which are high enough to overcompensate the electrical force.

2) Okay, we hopefully changed the statement now into the correct wording.

3) A completely grounded housing like in the DMA is not possible, since the high voltage contact has to be realized by a sliding contact from the outside. However, due to the rotation the device must be completely enclosed anyhow. Therefore, the applied high voltage to the outside does not impose any security issues.

Line 154: "radii" should read "radius"
Done.

Line 155: What do you mean with "particles already have a larger radius"?
Sorry, this was misleading, we tried to reformulate.
"Thus, if the particles are close to r_2, they are already located at a larger radius, compared to the centre radius of the aerosol inlet, when they enter the transfer zone, so they experience a higher centrifugal force."

Line 169: Figure caption of Figure 5: Are these "streamlines" or particle trajectories?
Yes, we changed it.

Line 173: What is meant with "…the transfer function of the two transfer functions…"?
This is a formulation error.

Line 177-182: Please describe your measurement setup and parameters in sufficient detail: e.g. what is the "previous" instrument? What DMA flow rates did you adjust? Which neutralizer was used (85-Kr or x-ray)? Which particle polarity did you classify?
We added some more information to that. Hopefully it's much clearer now.

In this case, a classifier (TSI 3080) with a DMA (TSI 3081) and a Kr-85 neutraliser (TSI 3077a) was used as the first instrument in the tandem setup, providing the mono-mobile aerosol. The voltage and desired volume flows are set there. To investigate the different operation parameters, $Q_a$ was set to $0.3\,\mathrm{lpm}$ or $1.5\,\mathrm{lpm}$ and $Q_{sh}$ was varied between $1.5\,\mathrm{lpm}$ and about $20\,\mathrm{lpm}$, resulting in different values for $\beta$. The second device was the CDMA, to which the same volume flows are applied via another classifier (TSI 3080). The used classifiers include a negative voltage power supply, so that both devices sample positively charged particles. To measure the transfer function, the measuring range of the CDMA is scanned step by step, i.e.

Line 186: "silver precipitates as nanoparticle": Do you mean that the silver vapor nucleates to form nanoparticles?
That is exactly what we meant. We changed it.

Line 190/191: The sentence, starting with "Because it takes approximately 30 minutes" doesn't make sense. Isn't it rather that the time it takes for a complete scan defines the requirement for the stability of the test aerosol?

The aim in this paper was to measure transfer functions. Measuring one transfer function takes about 30 mins. Measuring a whole distribution would take much more time, but consequently this should be the aim for the test aerosol. We reformulated.

Line 201/Figure 8: Every figure should generally -in conjunction with its caption- be self-explanatory. Figure 8b is far from this requirement. E.g. what are the "measurement data"? Axis caption (μ-1) and n₂/n₁ are also not described. Please modify the figure in a way that it is at least fundamentally (i.e. not in every detail) understandable without the need to read the text.
Edited and hopefully understandable, now.

[Figure]

(a) SEM image of the silver particles.  (b) Fitted Gaussian function to the measurement data.

**Figure 8.** SEM image of silver particles a) and ratio of measured number concentration at the exit of the second device of the tandem setup $n_2$ to number concentration at the inlet of the second device $n_1$, for different relative shifts of the CDMA transfer function to the fixed transfer function of the pre-classifying DMA b).

Line 272: The charge distribution of neutralized particles is more a convention rather than the ground truth. I would therefore always phrase that it is assumed to be known. Please add a reference for the charge distribution that you used (Wiedensohler approximation?)
Done.

Line 274: "simply" should read "singly"
Done.

Line 327 (Appendix B): $d_m$ and $d_{St}$ doesn't make much sense (see above), this should be $d_m$ and $d_{ae}$.
Hopefully this is clarified with the major concern at the beginning.

---

## Author Comment (AC2)

**Nomenclature**

[revised manuscript text omitted]

---

## Author Comment (AC4)

**2 Concept and fundamental theory**

The newly developed principle of CDMA combines the concepts of DMA and AAC. In general, the CDMA consists of two concentric cylinders between which high voltage can be applied, and/or both of which can be rotated at the same angular speed. This means that both the voltage and the speed can be superimposed, whereas in the DMA, only the voltage can be varied, and in the AAC, only the speed. This means that with the CDMA, particles can be classified according to their drag force and mass. In the DMA particles are usually characterized by their mobility diameter $d_m$ (i.e. the diameter of a spherical particle experiencing the same drag force for a given relative velocity as the actual particle) (Friedlander, 2000). The AAC typically uses the aerodynamic diameter $d_{ae}$ to characterize particles (i.e. the diameter of a sphere with unit density and the same settling velocity) which has the advantage that all particles of the same settling velocity in the centrifugal field will have the same equivalent diameter (Tavakoli and Olfert, 2014). However, particles with exactly the same shape but different material density will show different aerodynamic diameters. Since in our instrument the two-dimensional characterization mainly aims

2

to characterize the particle shape, we suggest to use the Stokes diameter $d_{St}$ instead (i.e. the diameter of a sphere of the same density with the identical settling velocity $w_S$ as the actual particle) for characterization (Colbeck, 2013; Reist, 1993). Although both equivalent sizes are closely related, the Stokes diameter does only depend on particle shape, e.g. characterized by the volume and mobility equivalent diameters $d_v$ and $d_m$, respectively (Baron et al., 2011):

$$d_{St} = \sqrt{\frac{18\eta}{\rho_S \cdot b \cdot Cu(d_m)} \cdot w_s} = \sqrt{\frac{\rho_0}{\rho_S} \cdot d_a e} = \sqrt{\frac{d_v^3}{d_m}} \tag{1}$$

With $\eta$ as the dynamic viscosity, $\rho_S$ the solid density of the particle material, $\rho_0$ unit density of $1000 \frac{\text{kg}}{\text{m}^3}$, $b$ centrifugal or gravitational acceleration, $w_S$ settling velocity. In particular, the Stokes and mobility diameter become identical in the case of a perfect sphere. However, the calculation of the Stokes diameter from classification according to settling velocity in the CDMA centrifugal field does require the knowledge of the particle density. Therefore, if the density is not known with sufficient accuracy or if an aerosol consisting of different materials is analyzed, the aerodynamic diameter should be used as in classical AAC theory. This can be easily adapted in the inversion algorithm. However, if the density of the particles is unknown, shape information is no longer accessible. By measuring all voltage-speed combinations, a full two-dimensional particle size distribution in terms of $d_{st}$ and $d_m$ can be calculated by data inversion.